# METALINT: GENERALIZABLE IDIOMATIC CODE QUALITY ANALYSIS THROUGH INSTRUCTION-FOLLOWING AND EASY-TO-HARD GENERALIZATION

## ABSTRACT

Large Language Models, though successful in code generation, struggle with code quality analysis because they are limited by static training data and can't easily adapt to evolving best practices. We introduce METALINT, an instruction-following framework that formulates code quality analysis as the task of detecting and fixing problematic semantic code fragments or code idioms based on high-level specifications. Unlike conventional approaches that train models on static code quality conventions, METALINT employs instruction tuning on synthetic linter-generated data with dynamic conventions to support easy-to-hard generalization, enabling models to adapt to novel or complex code patterns without retraining. To evaluate this, we construct a benchmark of challenging idioms inspired by real-world coding standards such as Python Enhancement Proposals (PEPs) and assess whether METALINT-trained models reason adaptively or simply memorize. Our results show that METALINT training improves generalization to unseen idioms. Qwen3-4B attains a 70.37% F-score on a manually curated and challenging PEP idiom detection benchmark, achieving the highest recall (70.43%) among all evaluated models. For localization, it reaches 26.73%, which is a strong outcome for its 4B parameter size and comparable to larger state-of-the-art models such as o3-mini, highlighting its potential for future-proof code quality analysis. Furthermore, METALINT training enables generalization in idiom detection across model families, model scales, synthetic data from diverse linters, and Java idioms, demonstrating the general applicability of our approach. We plan to release our code and data to enable reproducibility and further work.

## 1 INTRODUCTION

With the rise of Large Language Models (LLM) of code, concerns around the quality of generated code, such as readability, maintainability, efficiency, and security, have become increasingly prominent Singhal et al. (2024); Zheng et al. (2024). Researchers have been investigating the potential of LLMs to evaluate and improve code quality through benchmarks (Chambon et al., 2025; Singhal et al., 2024; Zheng et al., 2024; Waghjale et al., 2024), code review agents (Vijayvergiya et al., 2024; Rasheed et al., 2024), and static analysis with LLMs (Fang et al., 2025; Holden & Kahani, 2024; Khare et al., 2023). Several evaluation studies indicate that LLMs struggle with this task Singhal et al. (2024); Zheng et al. (2024), while attempts to improve them through prompting or training are limited by task-specific, static datasets often grounded in narrow or outdated coding practices (Vijayvergiya et al., 2024; Khare et al., 2023; Holden & Kahani, 2024; Zhang et al., 2024b). As a result, these systems often perform poorly when detecting rare issue types or when applied to code distributions that differ from their training data (Holden & Kahani, 2024). They may also over-flag outdated best practices, leading to a negative user experience and wasted time (Vijayvergiya et al., 2024). Ideally, we would develop LLM systems that can identify code quality issues without explicit supervision for target idioms—especially hard or rare patterns—and adapt to evolving best practices over time.

We approach this problem by training the LLM on a more general task: understanding and detecting semantic blocks of code, also known as *code idioms*. For example, a commonly used idiom for generating secrets or passwords in Python is to use the `random.choice` standard library

function. However, as noted in PEP 506 (D'Aprano, 2017), it is cryptographically insecure and Python documentation explicitly warns against using this module for security reasons, which is often missed by developers, as highlighted by accepted answers on forums like StackOverflow. PEP 506 also introduces a more secure semantic block or idiom in the form of the `secrets` module and the `secrets.choice` function, which acts as a safer alternative to the `random.choice` idiom. As illustrated by this example, detecting and locating idioms associated with bad practices can be leveraged for identifying code quality issues like *code smells* (Wikipedia contributors, 2024) or *Common Weakness Enumerations* (CWE) (MITRE Corporation, 2024). Additionally, these issues can be addressed by replacing instances of "bad" idioms with corresponding "good" idioms that align with best practices. Moreover, for this example and similar abstract idioms, constructing a precise rule-based approach is difficult. Simply flagging any use of `random.choice`, even in non-security-critical scenarios (e.g., randomization in a game engine), could result in a poor user experience. Vijayvergiya et al. (2024) show that LLMs can capture abstract notions of code quality, such as code idioms where building a linter or rule-based approach is challenging, by incorporating semantic reasoning about code and developer intent.

In this work, we train LLMs to recognize code idioms through a higher-level instruction-following task dubbed "meta-linting": given a specification of a best-practice code idiom $I$, the model learns to identify and localize non-idiomatic code fragments. Our pipeline is designed to support **easy-to-hard generalization** (Sun et al., 2024b). The easy cases involve simple idioms that can already be captured by existing linters, while the hard cases correspond to nuanced patterns such as PEP 506, where constructing precise rule-based checks is infeasible. To enable this, we generate synthetic training data for easy idioms using available linters and leverage it to improve performance on harder cases where linter support is lacking. While prior work such as Zhang et al. (2024c;b) has explored automated refactoring of non-idiomatic Python code, including the use of LLMs with prompting, our focus differs in three ways. First, we target challenging idioms beyond the reach of current linters. Second, we train on easy idioms with the goal of transferring detection ability to harder cases. Finally, we emphasize adaptability, aiming for LLMs that can accommodate evolving best practices provided in-context as instructions and examples, rather than memorizing a static rule sets.

To tackle meta-linting, we introduce METALINT, a training framework motivated by prior work showing that instruction tuning enables cross-task generalization and improves performance on unseen tasks (Mishra et al., 2021a; Sanh et al., 2021; Wang et al., 2022). Since meta-linting treats each idiom as a distinct task or code quality judgment, instruction fine-tuning (IFT) and preference optimization (PO) naturally extend detection ability to novel idioms. Existing linters (e.g., Ruff (ruf) for Python and PMD (pmd) for Java) provide large-scale synthetic data by enforcing simple idioms, which we use both for supervised IFT and as verifiers during PO to improve performance on harder idioms. To systematically study this generalization, we construct a benchmark of challenging idioms derived from popular PEPs introducing high-level constructs. We evaluate state-of-the-art reasoning and code models on this benchmark and compare them with METALINT trained models, examining whether they can move beyond memorizing easy idioms.

**Our key contributions are:**

1. We introduce METALINT, a training framework that leverages instruction following and synthetic data to enable easy-to-hard generalization while remaining adaptable to evolving best practices.
2. We construct a benchmark of challenging, broadly relevant code-quality idioms inspired by PEPs to evaluate the extent of easy-to-hard generalization achieved by METALINT.
3. We benchmark state-of-the-art code and reasoning models on our PEP hard-idiom benchmark and compare them against METALINT-trained models. Our method achieves the highest detection recall and competitive localization scores, even with smaller 4B models and without test-time compute.
4. We show that METALINT generalizes across programming languages (Python, Java), model families (Qwen, Llama), linters (Ruff, PMD, Tree-Sitter), test-time reasoning settings (with and without CoT), and model scales (3B–8B).

## 2 RELATED WORK

**Code Quality Analysis with Large Language Models.** A large body of prior work has explored the use of LLMs for code quality analysis through code review and static analysis. Tools like GPTLint (Travis Fischer, 2024) and lintrule (lin, 2023) treat LLMs as rule-guided linters via prompting or

fine-tuning. While Blyth et al. (2025) proposes a static analysis-driven prompting framework to improve LLM-generated code, Du et al. (2025) conversely uses LLMs to enhance static analysis tools by reducing false-positives. LintLLM (Fang et al., 2025) and (Shin et al., 2025) leverages LLMs for linting of Verilog and Quantum computing code. Khare et al. (2023) show LLMs outperform traditional static analysis tools for security-related CWEs with step-by-step reasoning. Vijayvergiya et al. (2024) train LLMs for best practice violation detection and localization, while Rasheed et al. (2024) design a multi-agent review pipeline for maintainability, efficiency, and bugs. Other works (Jiang et al., 2025b; Yao et al., 2025) use prefix-tuning and reinforcement learning with static analysis–based rewards for higher-quality, functionally correct code generation. Naik et al. (2024) and Jaoua et al. (2025) integrate LLMs with linters to produce more informative code reviews. RIdiom (Zhang et al., 2024c) introduces a rule-based way to identify and refactor non-idiomatic Python code with AST rewrite rules, while Zhang et al. (2024b) combines LLMs and rule-based detectors but doesn't explore nuanced idioms like PEP 506 or training LLMs to keep up with evolving best practices. Finally, CoUpJava (Jiang et al., 2025a) presents Java version upgrade benchmarks, conceptually similar to our hard PEP idiom benchmark for Python. Although prior work demonstrates the potential of LLMs for code quality tasks, it focuses on fixed rule sets or best practices that require retraining as they evolve. In contrast, we train models to interpret high-level specifications and perform static analysis, enabling broader generalization.

**Instruction Following for Generalization.** Instruction tuning has emerged as a powerful form of meta-learning that enables cross-task generalization by training models to interpret and follow natural language instructions rather than learning fixed tasks. Prior work shows diverse task instructions allow models to extract underlying task abstractions and apply them to unseen settings (Mishra et al., 2021b; Wang et al., 2022). Large-scale instruction tuning further improves zero- and few-shot generalization across tasks and modalities (Wei et al., 2021; Chung et al., 2022; Gao et al., 2021; Iyer et al., 2022; Brown et al., 2020). Instructions serve as high-density task representations, substituting supervision (Puri et al., 2022) and enabling generalization even with minimal labeled data or pseudo-labeled examples (Gu et al., 2022). Studies also show that instruction diversity drives generalization, with varied instructions outperforming repeated exposure to identical formats (Charton et al., 2024). This phenomenon holds across domains, including program synthesis where task-level prompting facilitates generalization in code generation models (Niu et al., 2023). SELF-GUIDE (Zhao et al., 2024) performs task-specific instruction following using synthetic data, demonstrating effectiveness, but relying entirely on LLM-generated data without verifiers. These results suggest instruction tuning acts as task-level meta-learning, enabling models to adapt to new tasks through natural language. Building on this we model specific code quality idioms as individual tasks and generate large-scale synthetic data for each meta-task to support cross-idiom generalization. This allows the trained model to keep pace with new idioms and evolving best practices. We also discuss additional related work on easy-to-hard generalization in Appendix B.

## 3 METHOD

We design the METALINT framework to teach an LLM to operationalize idiom descriptions provided in context, rather than memorizing specific idioms, thereby enabling adaptation to novel idioms at test time. We formulate idiom detection as an instruction-following *meta-task* $M_I$ for a given idiom $I$, where the prompt includes a natural language description $D_I$ and illustrative examples $E_I$, denoted as $M_I = \{D_I, E_I\}$. The LLM must identify all and only those code fragments that match idiom $I$ while performing $M_I$. This setup discourages rote memorization and encourages adaptive reasoning over the prompt's specification, since flagging violations of any other idiom $I' \neq I$ is penalized during $M_I$. By framing best practices as meta-tasks, this approach enables the LLM to remain flexible and better aligned with evolving best practices. We describe the components of our training framework in Figure 1 and Figure 2 below.

### 3.1 SYNTHETIC DATA GENERATION

One of the main goals of our meta-task formulation is enabling easy-to-hard generalization. We train LLMs on a set of "easy" idioms $I_{\mathcal{L}}$ that are detectable by existing linters $\mathcal{L}$, and evaluate them on a harder set $I_{\mathcal{L}'}$ consisting of idioms that linters cannot detect (where $\mathcal{L}'$ denotes the complement of $\mathcal{L}$, i.e., all idioms not detectable by a linter). Our hypothesis is that training on $I_{\mathcal{L}}$ helps the LLM

acquire the ability to understand and detect code idioms from in-context descriptions, enabling it to generalize more effectively to the harder idioms in $I_{\mathcal{L}'}$ compared to the untrained model. Since idioms in $I_{\mathcal{L}}$ are already covered by linters, we can leverage these tools to generate large-scale synthetic training data and provide supervision. For Python, we use the popular Ruff linter, which implements over 800 rules spanning syntax modernization, security, readability, etc., while for Java, we use the PMD static analyzer, which covers 269 idioms as well as some manually written tree-sitter[1] queries inspired by 8 Java Enhancement Protocols (JEPs) (Table 8). We run Ruff, PMD, and the JEP tree-sitter queries on Python and Java source code files $f \in \mathcal{F}$ from the STACK (Lozhkov et al., 2024) dataset, which contains code from a diverse range of GitHub repositories. This allows us to collect files with either no violations or one or more violations for each idiom in $I_{\mathcal{L}}$. Ruff also incorporates rules from other linters such as PyFlakes, Bandit, and autoPEP8, making it well-suited for producing diverse and representative synthetic data. Additionally, to automatically build the meta-task instruction prompts $M_{I_{\mathcal{L}}}$ for each idiom, we scrape rule-specific documentation from the Ruff and PMD websites, including descriptions and examples. An example prompt, along with a code file containing lines that violate the idiom, is shown in Appendix C.1. For the JEP tree-sitter queries, since they are few in number, we manually write the meta-task prompts.

## 3.2 INSTRUCTION SUPERVISED FINE-TUNING

As discussed in Section 3.1, we train the target LLM $\Phi$ on a set of linter-detectable, easy idioms $I_{\mathcal{L}}$, using the corresponding meta-task specifications $M_{I_{\mathcal{L}}}$ and a set of source code files $\mathcal{F}$. The input to the model consists of a prompt $p$, which combines a meta-task specification $M_I$ for some $I \in I_{\mathcal{L}}$ with a source code file $f \in \mathcal{F}$. The model's output is a list of idiom violations in the file, denoted as $V_{f,I}$, formatted as a JSON list with one violation per line (see example output in Appendix C.1). In cases where there are no violations ($|V_{f,I}| = 0$), the model is expected to output the phrase NO VIOLATIONS FOUND. We attempt to balance the data between positive (violations) and negative (no violations) examples as much as possible; however, due to the rarity of some Python idioms, the final distribution is approximately 70:30 in favor of files with no violations for Python data, but roughly 53:47 (PMD) and 50:50 (JEP Tree-Sitter) for the Java data. This results in a total of 53k synthetic training instances spanning 50 idioms (a subset of all the idioms detectable by Ruff) for Python Ruff data and 96.8k instances spanning 269 idioms for Java PMD and 127.3k instances spanning 15 idioms for tree-sitter data, respectively.

## 3.3 VERIFIABLE REWARD MODEL AND PREFERENCE OPTIMIZATION

For preference optimization, we adopt the RS-DPO approach (Khaki et al., 2024), which combines rejection sampling (RS) (Touvron et al., 2023) with Direct Preference Optimization (DPO) (Rafailov et al., 2023) to generate on-policy data from a supervised fine-tuned (SFT) policy model. It samples $k$ outputs per input, computes rewards for them, and constructs contrastive win–loss pairs based on the reward distribution and a threshold $\eta$ (Figure 2). We detail the verifiable linter-based reward model and contrastive pair sampling procedure below.

**Reward Model Design:** The reward model evaluates model outputs by comparing predicted violations against those flagged by the linter, treating the linter's line numbers (blue circle in "Verifiable Reward Model", Figure 1) as ground truth and the model's predicted lines (yellow circle) as predictions. Reward is computed using set-based precision, recall, and F1-score (visualized via the Venn diagram in the same figure), based on line-level overlap. Since each meta-task $M_I$ corresponds to a single idiom $I$, we compute one F1-score (reward) per instance.

**Sampling Contrastive Pairs:** We begin with an SFT policy model $\Phi^{SFT}$ and sample $k = 5$ outputs $y_i$, $i \in \{1, \ldots, k\}$ for each input $x$, using a range of temperature values $\tau = \{0, 0.3, 0.5, 0.7, 1.0\}$ to promote output diversity. Each response $y_i$ receives a reward $r_{y_i}$, and for each pair $(y_i, y_j)$, we compute the reward gap $|r_{y_i} - r_{y_j}|$. Pairs with a gap greater than the threshold $\eta = 0.2$ are added to the preference dataset $\mathcal{D}_p$. For any such pair where $r_{y_i} \geq r_{y_j} + \eta$, we assign $y_{win} = y_i$, $y_{lose} = y_j$, and store the instance $(x, y_{win}, y_{lose}) \in \mathcal{D}_p$. Following Khaki et al. (2024), we train the preference-tuned model $\Phi^{RL}$ using the DPO objective:

$$\Phi^{RL} = \arg\max \sum_{(x, y_{win}, y_{lose}) \in \mathcal{D}_p} \log \sigma \left( \beta \log \frac{\Phi^{RL}(y_{win}|x)}{\Phi^{SFT}(y_{win}|x)} - \beta \log \frac{\Phi^{RL}(y_{lose}|x)}{\Phi^{SFT}(y_{lose}|x)} \right)$$

---

[1]https://tree-sitter.github.io/tree-sitter/

Here, $\sigma$ denotes the sigmoid function, and $\beta = 0.1$ is the KL penalty coefficient, corresponding to low-to-moderate regularization.

### 3.4 TRAINING WITH REASONING TRACES

Finally, inspired by the success of reasoning-augmented models in code and math tasks, and their demonstrated effectiveness in improving CWE detection performance in LLMs (Khare et al., 2023), we propose SFT and DPO methods that incorporate chain-of-thought (CoT) reasoning. To obtain CoT traces that guide the LLM to correct answers, we adopt a rejection sampling approach ("Rejection Sampling SFT" in Figure 2) for SFT data collection. For each input $x$, we sample $k = 5$ responses $y_i, i \in \{1, \ldots, k\}$, from a base untrained CoT-capable LLM (e.g., Qwen3-4B), and compute a reward $r_{y_i}$ for each, following the RS-DPO procedure in Figure 2. Instead of forming contrastive pairs, we discard any $y_i$ with $r_{y_i} < \gamma$, where $\gamma = 1$, i.e., CoT–response pairs that are incorrect or improperly formatted. Rewards are applied only to the final response, obtained after parsing the CoT trace, and we also remove cases where the CoT fails to terminate or yield an answer. If no valid $y_i$ is found for an input $x$, we skip it. To promote meta-task diversity, we retain at most two valid responses per input: multiple $y_i$ only for violation cases and a single $y_i$ otherwise. This maintains the 71:29 no-violation-to-violation ratio of Ruff Python SFT data, with the latter more likely to fit within token limits. When excess valid responses exist, we keep the shortest completions, as they typically reflect more concise reasoning (final answers are of similar token length across samples). Following this policy, we collect 52.7k Python training instances from Ruff data, which we use to train the reasoning-enabled base Qwen3-4B with SFT. This yields a CoT-capable SFT model $\Phi_{CoT}^{SFT}$ for Python code quality analysis. We then apply the RS-DPO procedure in Section 3.3 and Figure 2, with the only change being that each $y_i$ now includes both the CoT trace and final response.

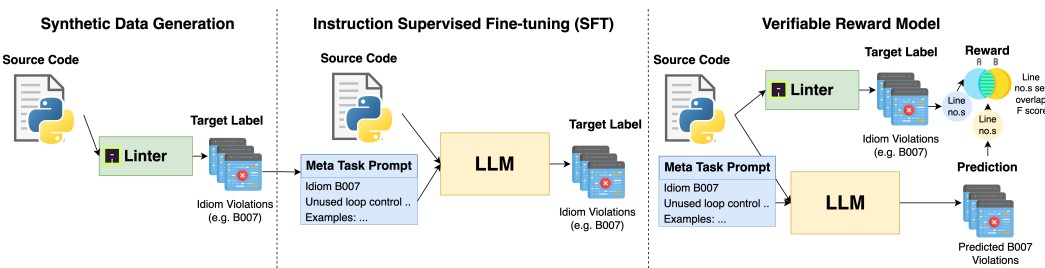

Figure 1: **METALINT:** (1) Synthetic data generation with linters/tools, (2) Supervised Instruction Fine-Tuning (SFT) on this data, and (3) Verifiable Reward Model derived from the linter.

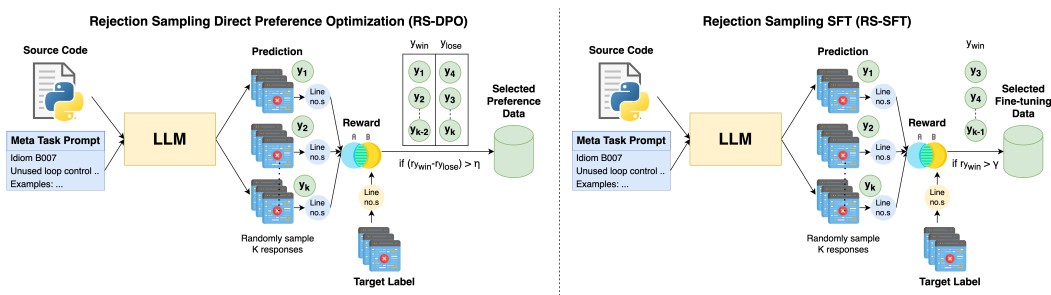

Figure 2: **METALINT:** Preference Optimization using reward model: (4) Rejection Sampling Direct Preference Optimization (RS-DPO), and (5) Rejection Sampling Supervised Fine-Tuning (RS-SFT).

## 4 EXPERIMENTS

### 4.1 EVALUATION METRICS

We evaluate the LLM's ability to detect idiom violations through two tasks: *detection*, which assesses whether a given idiom is violated in a code file, and *localization*, which evaluates whether

the model accurately identifies the specific line numbers where the violation occurs. For both tasks, we report precision, recall, and F-score metrics. Detection metrics are calculated at the corpus level for each idiom, treating each as a separate class, while localization metrics are computed at the instance level using set-based precision, recall, and F-score for the ground truth and predicted sets of violating line numbers. To handle potential class imbalance, we use macro-averaging across idioms and exclude `NO VIOLATION` as a class to penalize models that only predict `NO VIOLATIONS FOUND` (such models will score zero on all detection metrics). For localization, metrics are averaged only across instances with at least one line of idiom violation in the ground truth. Details of the formal definitions and exact computations of precision, recall, and F-scores for detection and localization are provided in Appendix D.1.

## 4.2 GENERALIZATION ON SYNTHETIC DATA

To evaluate whether METALINT training produces adaptive LLMs that handle evolving best practices and novel idioms at test time, we explore transfer settings spanning Python & Java.

**Ruff Python Idioms:** We construct a 5.3k-instance synthetic test set spanning 50 Ruff idioms, using the data generation procedure from Section 3.1. The data has a 74:26 no-violation-to-violation split, similar to the SFT training set. Idioms are chosen to vary in overlap with training idioms (Figure 3) and fall into three categories:

**In domain.** 5 idioms identical to those in SFT training, to assess whether METALINT improves performance on explicitly trained idioms.

**Near transfer.** 10 idioms with specifications similar but not identical to training idioms to probe memorization. Reliance on memorized patterns, may hurt performance due to interference.

**Far transfer.** 35 idioms distinct from training, to test whether the LLM can follow the provided specification and adapt to novel idioms at test time.

For these experiments, we use Qwen3-4B (with and without reasoning) and Llama-3.2-3B-Instruct to study the effect of test-time compute and model family.

**PMD and JEP Tree-Sitter Idioms:** For Java, we construct two synthetic test sets: 5.1k instances (54:46 split) spanning 269 PMD idioms, and 6.4k instances (50:50 split) spanning 15 JEP idioms (Table 5), flagged via tree-sitter queries. We evaluate in-domain performance by training the base LLM on the corresponding training set (Section 3.2), and also study transfer between PMD and JEP idioms to test adaptation to novel Java idioms. These experiments use Llama-3.2-3B-Instruct and Llama-3.1-8B-Instruct to assess the effect of model scale.

## 4.3 PEP HARD IDIOM BENCHMARK

**Benchmark Construction:** To test whether METALINT helps LLMs interpret high-level idiom specifications and generalize to nuanced idioms that linters miss, we construct a benchmark of "hard idioms" from 15 PEPs defining semantic or abstract behaviors beyond syntax. We design heuristics per PEP (Table 13, 14 and 15) to detect guideline violations and search the STACK-V2 corpus, prioritizing recall to retrieve broad candidate sets for manual selection. These idioms cannot be reliably detected by simple pattern matching, making them ideal for evaluating model's true understanding versus rote memorization. From the candidates, we handpick 15–20 representative files per PEP, and annotate the precise line ranges ("start" and "end") of the violated code, providing ground-truth for localization. We add negative examples for each PEP by picking files retrieved for a different PEP and making sure the current PEP is not violated, in order to have a balanced distribution of violation and no-violation cases. The final benchmark contains 536 examples (52% violations, 48% violation-free), enabling evaluation of METALINT's generalization from easy to hard Python idioms.

**Evaluating Easy-to-Hard Generalization:** We use the PEP hard idiom benchmark to test whether training on synthetic data for linter-detectable idioms improves performance on hard idioms. We evaluate the base model, SFT, and DPO-trained models on this benchmark.

**Benchmarking on Hard Idioms:** We evaluate state-of-the-art open and closed-source code and reasoning LLMs on the PEP hard idiom benchmark, comparing them to METALINT-trained models. Open-source models include instruction-tuned Qwen2.5 (Yang et al., 2024), Qwen2.5Coder (Hui et al., 2024), DeepSeek-R1-Distill-Qwen (DeepSeek-AI, 2025), Qwen3 (Team, 2025), and GPT-oss 20B/120B Agarwal et al. (2025). Closed-source models include GPT-4o (Hurst et al., 2024), o3 mini and o4 mini (OpenAI, 2025a), GPT-4.1 (OpenAI, 2025), and GPT-5 OpenAI (2025b). We select these models for their strong coding and reasoning performance and also evaluate the effects of code-specific pre-training, model scale (3B–120B), and test-time compute for open-source models.

## 5 RESULTS

To test whether MetaLint training leads to cross-idiom generalization instead of mere memorization of the training idioms and whether it can produce models that can keep up with evolving code quality standards, we present the transfer performance on the synthetic data for "easy" idioms in section 5.1. Then we explore the extent to which METALINT training achieves easy-to-hard generalization from the synthetic easy idioms to hard, manually curated PEP idioms in section 5.2. Finally, we compare METALINT trained models against state-of-the-art code and reasoning models on the manually curated hard PEP idioms in section 5.3.

### 5.1 GENERALIZATION ON SYNTHETIC DATA

**Python Ruff Idioms:** The performance of Qwen3-4B with and without reasoning and Llama3.2-3B-Instruct when trained on synthetic Ruff idioms and evaluated on the Ruff synthetic test set with varying transfer settings (section 4.2) is shown in Table 1 (full results in Table 18). While Table 1 shows the overall performance, we also analyze the performance broken down by each transfer setting in Table 16. The results show that the SFT stage leads to modest gains in detection and localization performance in most cases (except for a detection recall drop in the case of Llama3.2-3B-Instruct), but the DPO stage leads to huge gains in detection recall, F-score, and all localization metrics at the cost of a slight drop in detection precision. We identify that the drop in precision in the DPO stage is tightly controlled by the fraction of cases with no violations used in the DPO training and explore it in detail in Appendix D.3. Additionally, Table 16 shows that while SFT can lead to slight gains for the transfer settings (near transfer and far transfer), most gains emerge in the DPO stage, especially for non-reasoning models and detection recall. Overall this suggests that SFT can lead to memorization of the training idioms while DPO leads to generalization to novel idioms.

| Model | Detection | | | Localization | | |
|---|---|---|---|---|---|---|
| | $P_{Det}$ | $R_{Det}$ | $F_{Det}$ | $P_{Loc}$ | $R_{Loc}$ | $F_{Loc}$ |
| Qwen3-4B | 0.5380 | 0.2637 | 0.3539 | 0.1396 | 0.1479 | 0.1436 |
| Qwen3-4B + SFT | **0.7686** | 0.3178 | 0.4497 | 0.2976 | 0.2960 | 0.2968 |
| Qwen3-4B + SFT + RS-DPO | 0.7469 | **0.8315** | **0.7869** | **0.6527** | **0.6696** | **0.6611** |
| Qwen3-4B w CoT | 0.8812 | 0.6854 | 0.7710 | 0.5049 | 0.4878 | 0.4962 |
| Qwen3-4B w CoT + RS-SFT | **0.9350** | 0.8183 | 0.8727 | 0.6639 | 0.6500 | 0.6569 |
| Qwen3-4B w CoT + RS-SFT + RS-DPO | 0.9234 | **0.8643** | **0.8929** | **0.7710** | **0.7571** | **0.7640** |

Table 1: **Cross-Idiom Generalization on Python Ruff Idioms:** Effect of different METALINT training setups (SFT, RS-SFT, and RS-DPO) on Qwen3-4B (with and without reasoning). Best score across the compared training setups per model are bolded.

**PMD and JEP Tree-Sitter Idioms:** To demonstrate the generality of METALINT training across programming languages and linters, we present results from training on PMD and JEP Tree-Sitter synthetic data in Table 2 (full results in Table 28). Training on PMD shows the same overall pattern as before but with larger recall gains for both SFT and DPO, and notably stronger localization under DPO. For Llama3.1-8B-Instruct, SFT initially reduces detection precision, which DPO then recovers; the same precision dip-and-recovery appears when transferring PMD→JEP for Llama3.2-3B-Instruct. Despite never seeing JEP idioms during training, DPO models achieve strong detection and localization on JEP. In the untrained setting, Llama3.2-3B-Instruct (on PMD) and Llama3.1-3B-Instruct (on JEP) nearly always output the correct format but predict NO VIOLATIONS FOUND, yielding zero or near-zero scores because our metrics exclude that class for detection and only score positive cases for localization. Training on JEP yields high in-domain performance for all metrics with minimal additional benefit from DPO, likely due to JEP's smaller idiom set (15 vs 269 for PMD) and more precise instructions (Table 5). In the harder JEP→PMD transfer, DPO outperforms SFT, though overall transfer remains weaker than PMD→JEP, reflecting PMD's broader diversity and more challenging specifications (Appendix C.5).

Overall, METALINT training consistently yields more adaptable models than the base model, but

performance depends on the diversity of training idioms and the gap in instruction quality between training and test data.

| Model | Transfer | Detection | | | Localization | | |
|---|---|---|---|---|---|---|---|
| | | $P_{Det}$ | $R_{Det}$ | $F_{Det}$ | $P_{Det}$ | $R_{Det}$ | $F_{Det}$ |
| Llama3.2-3B-Instruct | | 0.0457 | 0.0079 | 0.0134 | 0.0015 | 0.0022 | 0.0017 |
| Llama3.2-3B-Instruct + SFT | PMD → PMD | 0.2251 | 0.4421 | 0.2983 | 0.2822 | 0.2778 | 0.2800 |
| Llama3.2-3B-Instruct + SFT + RS-DPO | | **0.4395** | **0.8908** | **0.5886** | **0.5930** | **0.5969** | **0.5949** |
| Llama3.2-3B-Instruct | | 0.3855 | 0.0096 | 0.0187 | 0.0005 | 0.0004 | 0.0005 |
| Llama3.2-3B-Instruct + SFT | PMD → JEP | 0.2286 | 0.4072 | 0.2928 | 0.1626 | 0.1336 | 0.1467 |
| Llama3.2-3B-Instruct + SFT + RS-DPO | | **0.4903** | **0.8338** | **0.6175** | **0.4216** | **0.3333** | **0.3721** |

Table 2: **Cross-Idiom Generalization on JEP & PMD Idioms:** Effect of different METALINT training setups (SFT and RS-DPO) on Llama3.2-3B-Instruct (Table 28). The transfer column indicates training and test data on the left and right side of the arrow. Best score across the compared training setups per model are bolded.

## 5.2 EVALUATING EASY-TO-HARD GENERALIZATIONS

To evaluate whether METALINT training on easy, linter-detectable Ruff idioms improves performance on hard, manually curated PEP idioms, we report results on our PEP hard idiom benchmark (Table 3, full results in Table 19). At the SFT stage, performance declines for Qwen3-4B (with and without CoT) but improves slightly for Llama3.2-3B-Instruct, suggesting that SFT can induce memorization of the training distribution and reduce adaptability. In contrast, DPO yields clear improvements in detection and localization (except detection precision for Llama3.2-3B-Instruct), with statistically significant gains (Appendix E.2). An additional experiment training Qwen3-4B (CoT) directly with RS-DPO, bypassing SFT, resulted in near-zero performance because many generated DPO pairs violated the required output format, which the model inherited. Thus, SFT, despite its drawbacks, is essential for teaching format compliance and setting the stage for DPO to unlock easy-to-hard generalization. Interestingly, the non-CoT model achieves substantially higher detection recall and slightly higher F-score than the CoT variant, despite lower precision. Our analysis attributes the CoT model's reduced recall to its more conservative interpretation of idiom specifications and to errors such as misinterpretation, overthinking, and skipped lines, as detailed in Appendix E.3.

| Model | Detection | | | Localization | | |
|---|---|---|---|---|---|---|
| | $P_{Det}$ | $R_{Det}$ | $F_{Det}$ | $P_{Loc}$ | $R_{Loc}$ | $F_{Loc}$ |
| Qwen3-4B | 0.5267 | 0.1715 | 0.2587 | 0.0954 | 0.0824 | 0.0884 |
| Qwen3-4B + SFT | 0.4333 | 0.0821 | 0.1381 | 0.0432 | 0.0221 | 0.0292 |
| Qwen3-4B + SFT + RS-DPO | **0.7031** | **0.7043** | **0.7037** | **0.3536** | **0.1930** | **0.2497** |
| Qwen3-4B w CoT | 0.8154 | 0.3986 | 0.5354 | 0.2625 | 0.1467 | 0.1882 |
| Qwen3-4B w CoT + RS-SFT | 0.7615 | 0.3689 | 0.4970 | 0.2785 | 0.1437 | 0.1896 |
| Qwen3-4B w CoT + RS-SFT + RS-DPO | **0.9303** | **0.4958** | **0.6468** | **0.3482** | **0.2169** | **0.2673** |

Table 3: **Easy-to-Hard Generalization on PEP Idioms:** We evaluate the effect of different METALINT training setups (SFT, RS-SFT, and RS-DPO) on Qwen3-4B (with and without reasoning) and Llama3.2-3B. Models are trained on easy synthetic Python Ruff idioms and tested on hard manually curated PEP idiom detection data which can't be handled by linters or static analyzers (section 4.3). Best score across the compared training setups per model are bolded.

## 5.3 BENCHMARKING ON HARD IDIOMS

Table 4 compares the best-performing Qwen3-4B METALINT DPO models against state-of-the-art code and reasoning models (full results in Table 17).

**Detection:** In terms of detection F-score, the non-CoT METALINT model is competitive with o3-mini and GPT-5 but is outperformed by some larger open-source models (e.g., Qwen3-32B with

CoT, DeepSeek-R1-Distill-Qwen-32B with CoT, and GPT-oss-120B) and closed-source models (GPT-4o, GPT-4.1, and o4-mini). However, the non-CoT model achieves the highest detection recall among all evaluated models, while the CoT model ranks among the top in precision, surpassed only by Qwen3-32B with CoT and o4-mini.

**Localization:** For localization, the METALINT models lag behind larger 32B and 120B models (such as Qwen3-32B, Qwen2.5Coder-32B, and DeepSeek-R1-Distill-Qwen-32B) and the GPT models, but perform comparably to o3-mini (statistical significance analysis in Appendix E.2) and outperform GPT-oss-20B. This is notable given that the METALINT models are much smaller (4B parameters), trained only on synthetic data derived from easy idioms, and that the non-CoT model does not use test-time compute.

Overall, the strong results, especially the best-in-class recall of the non-CoT model, demonstrate the effectiveness of our framework in achieving easy-to-hard generalization. This is enabled by training on synthetic data with easy idioms and by encouraging adaptive reasoning through instruction fine-tuning and DPO rather than relying on rote memorization.

| Model | Detection | | | Localization | | |
|---|---|---|---|---|---|---|
| | $P_{Det}$ | $R_{Det}$ | $F_{Det}$ | $P_{Loc}$ | $R_{Loc}$ | $F_{Loc}$ |
| Qwen3-8B | 0.8267 | 0.3572 | 0.4988 | 0.1806 | 0.1285 | 0.1501 |
| Qwen3-8B with CoT | 0.8886 | 0.4672 | 0.6124 | 0.3122 | 0.2029 | 0.2459 |
| Qwen3-14B | 0.9021 | 0.4612 | 0.6103 | 0.2890 | 0.2521 | 0.2693 |
| Qwen3-14B with CoT | 0.9116 | 0.4857 | 0.6337 | 0.3993 | 0.2915 | 0.3369 |
| Qwen3-32B | 0.9021 | 0.5205 | 0.6601 | 0.2807 | 0.2711 | 0.2758 |
| Qwen3-32B with CoT | 0.9377 | 0.5645 | 0.7048 | 0.4152 | 0.3086 | 0.3540 |
| Qwen2.5-32B-Instruct | 0.8667 | 0.2656 | 0.4066 | 0.1630 | 0.1477 | 0.1550 |
| Qwen2.5Coder-32B-Instruct | 0.8961 | 0.5328 | 0.6683 | 0.3432 | 0.3077 | 0.3245 |
| DeepSeek-R1-Distill-Qwen-32B with CoT | 0.9008 | 0.5899 | 0.7130 | 0.4015 | 0.3403 | 0.3684 |
| GPT-oss-20b | 0.8377 | 0.3531 | 0.4968 | 0.2510 | 0.1695 | 0.2024 |
| GPT-oss-120b | 0.9157 | 0.6456 | 0.7573 | 0.3991 | 0.3331 | 0.3631 |
| **Qwen3-4B METALINT (SFT+RS-DPO)** | 0.7031 | **0.7043** | 0.7037 | 0.3536 | 0.1930 | 0.2497 |
| **Qwen3-4B METALINT w CoT (RS-SFT + RS-DPO)** | 0.9303 | 0.4958 | 0.6468 | 0.3482 | 0.2169 | 0.2673 |
| o3-mini | 0.8939 | 0.5845 | 0.7068 | 0.3169 | 0.2361 | 0.2706 |
| o4-mini | **0.9667** | 0.5943 | 0.7361 | 0.4131 | 0.3164 | 0.3584 |
| GPT-4o | 0.8938 | 0.6788 | **0.7716** | 0.4461 | 0.3320 | 0.3807 |
| GPT-4.1 | 0.9070 | 0.6460 | 0.7546 | **0.4632** | **0.4673** | **0.4653** |
| GPT-5 (high) | 0.9130 | 0.5673 | 0.6998 | 0.4397 | 0.4257 | 0.4326 |

Table 4: **Benchmarking on Hard Idioms:** Results comparing state of the art code and reasoning models on the hard PEP benchmark to contextualize the gains achieved with METALINT training. The best scores are bolded and second best and underlined.

## 6 CONCLUSION AND FUTURE WORK

Our results show that METALINT training fosters adaptive reasoning over idiom specifications rather than rote memorization. We observe generalization to unseen idioms in Python and Java, across three linters (Ruff, PMD, JEP tree-sitter), two model families (Qwen, Llama), reasoning and non-reasoning settings, and multiple scales (3B, 4B, 8B). Easy-to-hard generalization occurs from linter-detectable Ruff idioms to harder PEP idioms, with SFT teaching output formatting and DPO enabling true generalization. Compared to state-of-the-art code and reasoning models, METALINT-trained Qwen models have detection comparable with o3-mini and GPT-5, achieving highest recall (non-CoT) and third-best precision (CoT). Localization lags but surpasses GPT-oss-20B with only 4B parameters and no test-time compute and is comparable to o3-mini, demonstrating efficiency. These results highlight the effectiveness of instruction fine-tuning and preference optimization on synthetic data for reasoning and generalization, even with scarce annotated examples. For mechanically easy idioms, linters remain cost-effective, but METALINT enables detection of abstract idioms, supporting personalized, evolving code quality standards. We plan to release code and data for reproducibility. Future work includes training for automated refactoring and exploring advanced RL methods like Group Relative Policy Optimization (GRPO) Shao et al. (2024).

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

## A    LIMITATIONS

Despite the promising results achieved by METALINT, our work has some limitations that we plan to address in future research. For the CoT setting, we didn't explore whether non-CoT models can be trained to effectively produce CoT-style reasoning with supervision from a teacher model. We also explored self-improvement strategies for RS-SFT data generation in cases where the base model failed, such as STaR (Zelikman et al., 2022), but found it challenging to generate CoTs that do not directly reference provided hints, which risks contaminating the training data. As a result, we adopted a simpler rejection sampling or RS-SFT strategy. Furthermore, our approach does not yet incorporate more advanced reinforcement learning techniques such as Group Relative Policy Optimization (GRPO) (Shao et al., 2024) using our verifiable linter-based reward model, or curriculum learning methods to control the progression of idiom difficulty within synthetic training data. Our current experiments also focus on training on one language at a time, such as only Python or Java. Future work will explore joint training and extension to more programming languages like JavaScript, Ruby, Go, etc., as well as cross-language generalization by training on Python idioms and evaluating on Java idioms, and vice versa. Finally, while we do not evaluate or train for refactoring of the idiom-violating code, we plan to do so in future work.

## B    MORE RELATED WORK

**Easy-to-Hard Generalization.** Research shows that training on simpler problems enhances generalization to harder ones in math, algorithms, and code, motivating its application to code quality analysis. In math reasoning, models trained on easier problems (e.g., level 1–3) consistently generalize better to harder benchmarks (e.g., level 4–5) (Bai et al., 2024; Shafayat et al., 2025; Parashar et al., 2025). Several works emphasize the importance of selecting high-quality supervision for

harder problems (He et al., 2024). Beyond math, Sun et al. (2024a) shows that reward models trained on simple code and math problems improve performance on complex ones. Broader studies on multi-task and length generalization (Hu et al., 2025) and differentiable programming (Gaunt et al., 2016) reveal how structural simplicity during training can lead to robustness on longer or more complex reasoning instances, including code. Zhang et al. (2024a) reinforces this by evaluating reward models on algorithmic tasks like string manipulation and demonstrating transfer from simpler to harder formats. Drawing inspiration from this work, we train METALINT on large-scale synthetic data covering easily detectable code idioms handled by rule-based linters, and hypothesize that these simple patterns serve as stepping stones toward generalizing to complex, novel PEP idioms.

## C    METHOD ADDITIONAL DETAILS

### C.1    METALINT INSTRUCTION FOLLOWING PROMPT

We used the following instruction following style prompt to train the model with synthetic Ruff idiom data for the meta-linting task:

---

**METALINT Instruction Following Prompt**

Look at the following list of code idiom specifications with definitions and examples: {LIST_OF_IDIOM_SPECS}

Given these idioms, your task is to look at a code file and detect violations of the above idioms, and flag them like a linter. You should also suggest a fix if possible. Report the results per idiom specification mentioned above and just say `NO VIOLATIONS FOUND` if no violations are found for a given idiom. Do not detect any idioms not specified above.

Code file: {CODE_FILE}

Violations per idiom:

---

An example input with the code file and idiom spec populated as well as the expected JSON style output is shown below:

---

**Example Ruff Meta-Task Input**

Look at the following list of code idiom specifications with definitions and examples: #
Idiom ANN202 (missing-return-type-private-function)
Definition: Checks that private functions and methods have return type annotations.
Rationale: Type annotations are a good way to document the return types of functions. They
also help catch bugs, when used alongside a type checker, by ensuring that the types of any
returned values, and the types expected by callers, match expectation.
Example:

```
def _add(a, b):
    return a + b
```

Use instead:

```
def _add(a: int, b: int) -> int:
    return a + b
```

Given these idioms, your task is to look at a code file and detect violations of the above
idioms, and flag them like a linter. You should also suggest a fix if possible. Report the
results per idiom specification mentioned above and just say 'NO VIOLATIONS FOUND'
if no violations are found for a given idiom. Do not detect any idioms not specified above.
Code file:

```
  1 # -*- coding: utf-8 -*-
  2 # pragma pylint: disable=unused-argument, no-self-use
...
 86     def _reload(self, event, opts):
 87         """Configuration options have changed,
    save new values"""
 88         self.options = opts.get("fn_cisco_amp4ep", {})
 89         validate_opts(self)
 90
 91     @function("fn_amp_move_computer")
 92     def _fn_amp_move_computer_function(self, event, *args,
    **kwargs):
 93         """Function: Move computer to a group with given
    connector guid and group guid."""
 94         try:
...
```

Violations per idiom:

---

**Example Ruff Meta-Task Output**

**Idiom ANN202 Violations:**

```
{"line": " 86     def _reload(self, event, opts):", "fix": null}
{"line": " 92     def _fn_amp_move_computer_function(self,
event, *args, **kwargs):", "fix": null}
```

---

## C.2 DPO CONTRASTIVE PAIR AND RS-SFT SAMPLING DETAILS

To generate RS-DPO contrastive samples (or RS-SFT outputs) from the baseline SFT (or untrained)
models, we used the following hyperparameters: nucleus sampling with a maximum of 2048 new
tokens, $k = 5$ sampled outputs per input, temperatures picked cyclically from $\{0, 0.3, 0.5, 0.7, 1\}$,
a top-$p$ (cumulative probability threshold) of 0.95, and a seed of $42 + i$, where $i \in \{1, \ldots, k\}$, to
encourage both reproducibility and output diversity.

For RS-DPO sampling (in both CoT and non-CoT settings), we used the standard METALINT
instruction-following prompt with the SFT models. In contrast, for RS-SFT output sampling from

the untrained model, we employed the expanded "Baseline Inference Prompt" described in Section C.4.

## C.3 Training Hyperparameters and Computational Environment

**Python SFT/RS-SFT hyperparameters:**
We fine-tune the `Qwen3-4B` model using `flash_attention_2` and `bfloat16` precision. The model is trained for 2 epochs with a learning rate of 2e-5, cosine learning rate schedule, and a warmup ratio of 0.1. We use a maximum sequence length of 3000 tokens, a per-device batch size of 2, and gradient accumulation steps of 4. Gradient checkpointing is enabled to reduce memory usage, with non-reentrant mode. Evaluation is performed every 2000 steps, and checkpoints are saved at the same interval. Special tokens are manually handled in the chat template without automatic insertion. The training uses 12 preprocessing workers and is seeded with 42 for reproducibility.

**Python RS-DPO parameters:**
We fine-tune the model using RS-DPO with `bfloat16` precision and a reward shaping parameter $\beta = 0.1$. Training is performed for 1 epoch with a learning rate of 5e-7, cosine learning rate scheduling, and a warmup ratio of 0.1. We use a maximum input length of 3500 tokens, a per-device batch size of 2, and gradient accumulation steps of 4. Gradient checkpointing is enabled with non-reentrant mode to optimize memory usage. The optimizer is `AdamW`, and evaluation is conducted every 200 steps with checkpoints saved at the same interval. The training is seeded with 42 for reproducibility.

**Java SFT hyperparameters:**
For Java experiments, we fine-tune `Llama-3.1-8B-Instruct` and `Llama-3.2-3B-Instruct` with `bfloat16` precision. Both models are trained for 2 epochs with a learning rate of 2e-5, cosine learning rate schedule, and warmup ratio of 0.1. We use a maximum sequence length of 3000 tokens, per-device batch size of 2, and gradient accumulation steps of 4. Gradient checkpointing (non-reentrant) is enabled. Evaluation and checkpoint saving occur every 5000 steps. Special tokens are manually handled in the chat template. Training is seeded with 42.

**Java RS-DPO parameters:**
RS-DPO training is performed on `Llama-3.1-8B-Instruct` and `Llama-3.2-3B-Instruct` using `bfloat16` precision. Training runs for 1 epoch with a learning rate of 5e-7, cosine learning rate scheduling, and warmup ratio of 0.1. We use a maximum input length of 3500 tokens, a per-device batch size of 2, and gradient accumulation steps of 4. Gradient checkpointing (non-reentrant) is enabled. Evaluation and checkpoints are recorded every 200 steps. Reward shaping parameters vary across settings, with $\beta \in \{0.1, 0.5, 1\}$. Seeds are fixed at 42 for reproducibility.

**Computational Environment:**
All SFT, RS-SFT, and RS-DPO experiments (Python and Java) were conducted on a Linux server equipped with NVIDIA A100 80GB GPUs (Ampere architecture), CUDA 12.9, and driver version 575.51.03. Each job had access to 100 GB of CPU memory and 2 CPU cores. Training used mixed-precision (`bfloat16`) with gradient checkpointing to optimize memory usage. Inference used a similar setup with GPU allocation varying by model size.

## C.4 Baseline Inference Details

We use the following hyperparameters for performing inference with the baseline LLMs:
**Open Source LLMs:** We perform nucleus sampling with 8192 max-new tokens, temperature of 0.7, top-p (cumulative probability threshold) of 0.95 and seed of 42 (to promote reproducibility).
**Closed Source LLMs:** We use the chat completion OpenAI API with max tokens of 1024 for GPT-4.1 and GPT-4o and max completion tokens of 3000 for o3-mini and o4-mini. We use default parameters for everything else (temperature of 1 and top-p of 1, no presence penalty). For GPT-5 we use 8192 max completion tokens and high reasoning effort.

Additionally, we use an expanded prompt (Baseline Inference Prompt) compared to the one used for METALINT, specifically adding more details about output formatting to ensure all baselines have a fair chance and do not suffer performance drops due to formatting mismatches. For the same

reason, we also allow certain relaxations in output formatting during evaluation on the PEP Hard Idiom Benchmark.

---

**Baseline Inference Prompt**

Look at the following list of code idiom specifications with definitions and examples: {LIST_OF_IDIOM_SPECS}

Given these idioms, your task is to look at a code file and detect violations of the above idioms, and flag them like a linter. You should also suggest a fix if possible. Report the results per idiom specification mentioned above and just say `NO VIOLATIONS FOUND` if no violations are found for a given idiom. Do not detect any idioms not specified above.

Code file: {CODE_FILE}

# OUTPUT FORMAT

I want you to generate your output under a section called "### Final Idiom Violations Found".

Structure you response for a given idiom XYZ as follows for cases with violations:

### Final Idiom Violations Found

**Idiom XYZ Violations:**

```
{"line": " 12 \\t\\t#event = forms.ModelChoiceField(queryset=
Inquiry.objects.filter(owner=kwargs.pop('user')))", "fix": null}
{"line": "  1 from django import forms\\n
2 from django.forms.models import inlineformset_factory\\n
3 from .models import Request\\n
4 from inquiry.models import *",
"fix": [{"before": "from django import forms\\n
from django.forms.models import inlineformset_factory\\n
from .models import Request\\n
from inquiry.models import *\\n\\n\\n\\n",
"after": "from django import forms\\n
from django.forms.models import inlineformset_factory\\n
from inquiry.models import *\\n\\n
from .models import Request\\n\\n\\n"}]}
```

and as follows for cases with violations:

### Final Idiom Violations Found

**Idiom XYZ Violations:**

```
NO VIOLATIONS FOUND
```

Violations per idiom:

---

## C.5 PMD IDIOM SPECIFICATIONS

We scrape PMD idioms specification from the Java section of the PMD rules documentation `https://docs.pmd-code.org/latest/pmd_rules_java.html`. The PMD instructions are more complex and more ambiguous than our handcrafted JEP specifications because the examples are more verbose and don't pinpoint the specific lines that should be flagged as idiom violations, as can be seen in the example below.

**PMD Rule Specification:** UnitTestShouldIncludeAssert

```
Since: PMD 2.0
Priority: Medium (3)
Unit tests should include at least one assertion. This makes
the tests more robust, and using assert with messages provide
the developer a clearer idea of what the test does. This rule
checks for JUnit (3, 4 and 5) and TestNG Tests. Note: This rule
was named JUnitTestsShouldIncludeAssert before PMD 7.7.0. This
rule is defined by the following Java class:
net.sourceforge.pmd.lang.java.rule.bestpractices.
UnitTestShouldIncludeAssertRule

Example(s):
public class Foo {
    @Test
    public void testSomething() {
        Bar b = findBar();
        // This is better than having a NullPointerException
        // assertNotNull("bar not found", b);
        b.work();
    }
}

This rule has the following properties:

Name
Default Value
Description

extraAssertMethodNames

Extra valid assertion methods names

Use this rule with the default properties by just referencing
it:
<rule ref="category/java/bestpractices.xml/
UnitTestShouldIncludeAssert" />

Use this rule and customize it:
<rule ref="category/java/bestpractices.xml/
UnitTestShouldIncludeAssert">
    <properties>
        <property name="extraAssertMethodNames" value="" />
    </properties>
</rule>
```

---

**Java METALINT Instruction Following Prompt**

**Task Instructions (1/2):**
Look at the following code idiom specification with definitions and examples:
{IDIOM_SPEC}

**Task Instructions (2/2):**
Given this idiom, your task is to look at a code file and detect violations of the above idiom, and flag them like a linter. You should also suggest a fix if possible. Report the results for only the idiom specification mentioned above and just say `NO VIOLATIONS FOUND` if no violations are found for the given idiom. Do not detect violations of any idiom not specified above.

**Code file:**
{CODE_FILE}

**Violations per idiom:**

---

# D ADDITIONAL EXPERIMENTAL DETAILS

## D.1 EVALUATION METRICS

Let $I$ denote an idiom, $M_I$ its corresponding meta task specification, $f \in \mathcal{F}$ a code file, $V_{f,I}$ the ground truth set of violating line numbers, and $\hat{y} = V_{f,I}^{\Phi}$ the model predicted violations. For each dataset instance with input prompt $x$ and ground truth set of line numbers $y$, $(x, y) = (\{f, M_I\}, V_{f,I}) \in \mathcal{D}$.

We define the indicator variable:

$$\mathbb{1}[x] = \begin{cases} 1 & \text{if } x \text{ is true} \\ 0 & \text{otherwise} \end{cases}$$

**Detection Metrics:**

$$P_I = \frac{\sum_{(x,y)\in\mathcal{D}} \mathbb{1}[|y| > 0] \cdot \mathbb{1}[|\hat{y}| > 0]}{\sum_{(x,y)\in\mathcal{D}} (\mathbb{1}[|y| > 0] \cdot \mathbb{1}[|\hat{y}| > 0] + \mathbb{1}[|y| = 0] \cdot \mathbb{1}[|\hat{y}| > 0])}$$

$$R_I = \frac{\sum_{(x,y)\in\mathcal{D}} \mathbb{1}[|y| > 0] \cdot \mathbb{1}[|\hat{y}| > 0]}{\sum_{(x,y)\in\mathcal{D}} (\mathbb{1}[|y| > 0] \cdot \mathbb{1}[|\hat{y}| > 0] + \mathbb{1}[|y| > 0] \cdot \mathbb{1}[|\hat{y}| = 0])}$$

Macro-averaged detection metrics:

$$P_{\text{Det}} = \frac{1}{|I|} \sum_I P_I, \quad R_{\text{Det}} = \frac{1}{|I|} \sum_I R_I, \quad F_{\text{Det}} = \frac{2P_{\text{Det}}R_{\text{Det}}}{P_{\text{Det}} + R_{\text{Det}}}$$

**Localization Metrics:**

$$P_{\text{Loc}} = \frac{1}{|\mathcal{D}|} \sum_{(x,y)\in\mathcal{D}} \frac{|y \cap \hat{y}|}{|\hat{y}|}, \quad R_{\text{Loc}} = \frac{1}{|\mathcal{D}|} \sum_{(x,y)\in\mathcal{D}} \frac{|y \cap \hat{y}|}{|y|}, \quad F_{\text{Loc}} = \frac{2P_{\text{Loc}}R_{\text{Loc}}}{P_{\text{Loc}} + R_{\text{Loc}}}$$

## D.2 IDIOMS CHOSEN FOR RUFF IDIOM TRANSFER DATASET

Table 9 lists the Ruff idioms used in the SFT training and synthetic transfer evaluation test sets. Idioms are grouped by their source linter and cover a range of syntax, semantics, naming, and upgrade-related rules.

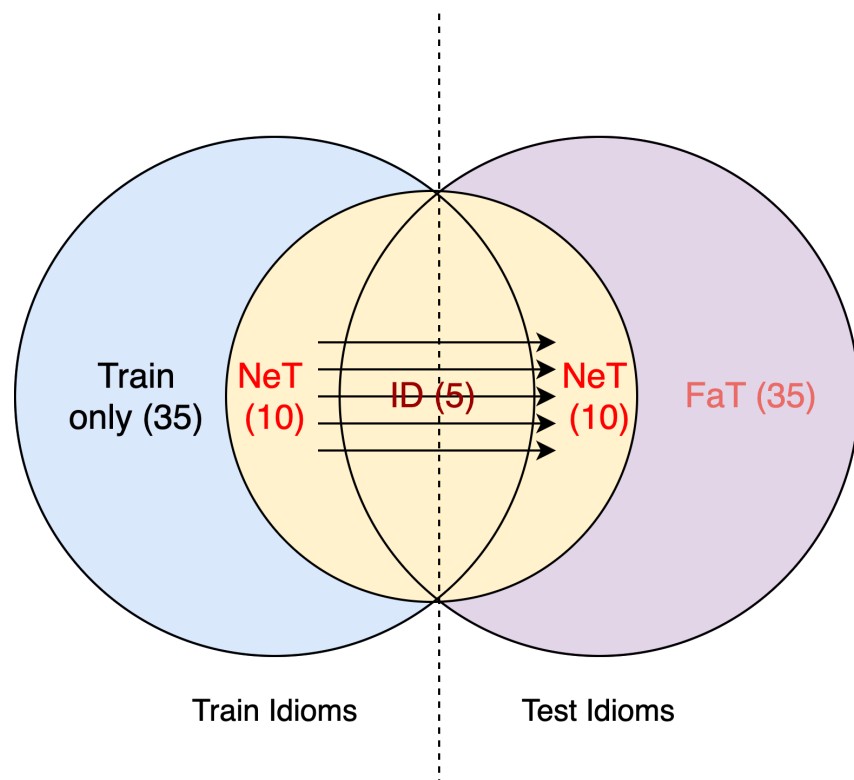

Figure 3: ID: In-Domain, NeT: Near Transfer, FaT: Far Transfer.

### D.3 DPO NO VIOLATION FRACTION ABLATIONS

We analyze the impact of varying the amount of samples with zero violations used for RS-DPO training. These experiments were motivated by initial findings comparing models trained only on data with at least one violation to those trained on the full dataset. By design, RS-DPO generates significantly more training data for cases with at least one violation, due to greater variance in reward signals. This is further amplified by the fact that the initial SFT policy/checkpoint is already quite accurate in handling cases with NO VIOLATIONS FOUND leading to low variance in reward across responses.

Our early experiments showed that excluding all NO VIOLATIONS FOUND cases led to notable gains in recall and line-level localization. However, this came at the cost of a significant drop in precision compared to the SFT policy/base model. Further analysis revealed a sharp decline in the accuracy of predicting NO VIOLATIONS FOUND, from nearly 99% down to 70-80%, with performance worsening monotonically over training steps. Conversely, training on the full dataset (i.e., including 100% of the NO VIOLATIONS FOUND cases) improved precision but offered only modest gains in recall and localization, which also degraded with continued training. These findings suggest that while some NO VIOLATIONS FOUND data is necessary to maintain high precision, too much of it may hinder recall and localization.

To investigate this trade-off, we experimented with keeping only a fraction of the NO VIOLATIONS FOUND data during training. Specifically, we randomly sampled $k\%$ of such data, varying $k$ across $\{0\%, 2\%, 5\%, 10\%, 20\%, 40\%, 100\%\}$. These percentages were selected based on observed trends: 20%, 40%, and 100% yielded similar results, which discouraged further tests at 60% or 80%, while 2% and 5% were chosen due to a noticeable performance jump between 0% and 10%. We found that 5% offered a favorable middle ground, largely retaining or slightly reducing precision, while preserving most of the recall (resulting in the highest detection F-score), and only modestly impacting line-level localization. Based on these insights, we conducted a limited ablation on the CoT model,

evaluating 2% and 5% inclusion to determine the optimal setting for both detection and localization (as shown in Table 11).

### D.4 PEP Benchmark Creation Additional Details

As discussed in section 4.3 we use some high recall heuristics to find promising candidates for detecting the selected hard PEP idioms. These are summarized in Table 13, 14 and 15.

## E More Results

### E.1 Expanded Results on the PEP Hard Idiom Benchmark

We show the expanded results across various model sizes for the evaluated model families in Table 17. We note that most results follow the expected trends with more parameters or CoT usage leading to better performance but there are soem exceptions to the trend. We mainly see this for cases like Qwen2.5 and Qwen2.5Coder families. We note that Qwen2.5Coder-7B-Instruct has almost zero metrics because it always predicts `NO VIOLATIONS FOUND` for all instances and Qwen2.5Coder-14B-Instruct has really low scores because of similar reasons. for Qwen2.5 family we notice that 32B variant performs a bit worse than 32B.

We also analyze METALINT SFT models on the hard PEP benchmark and observe that they perform similarly or slightly worse than the base untrained models. This suggests that SFT alone may lead to overfitting on the Ruff idiom distribution and struggles to generalize from easy to hard cases without DPO training. These findings highlight the importance of the DPO (preference-tuning) stage in the METALINT pipeline. However, we also emphasize that while the SFT stage can limit generalization, it remains essential for effective DPO training, as it teaches the LLM to follow the correct output format and establishes a strong base policy. This is supported by our experiments with the CoT model, where applying RS-DPO directly to the Qwen/Qwen3-4B model (without SFT) led to near-zero performance across all metrics, as the model consistently failed to produce outputs in the required format.

### E.2 Statistical Significance of Results on the PEP Hard Idiom Benchmark

To analyze the statistical significance of performance differences over the PEP benchmark, we conduct Wilcoxon signed-rank tests comparing various METALINT variants against each other and against baseline models. We evaluate instance-level detection accuracy (binary labels indicating whether the LLM correctly predicted the presence of a violation) as well as instance-level precision and recall for line-level localization. To control for multiple comparisons, we apply a Bonferroni correction to adjust the significance threshold $\alpha$ as $\alpha = \frac{0.05}{m}$ where $m$ is the number of comparisons (or rows in any given statistical significance table in this case).

Table 21 reports the Wilcoxon signed-rank test statistic and corresponding $p$-value (in parentheses) for detection accuracy, localization precision, and localization recall when comparing various METALINT variants to assess the effects of RS-DPO and CoT. We find that applying RS-DPO to the base SFT policy leads to statistically significant improvements in both detection and localization performance, with RS-DPO consistently outperforming the original SFT checkpoint across all three metrics with it being always better for localization. For the CoT variant, RS-DPO also yields consistent but less significant gains, likely because the RS-SFT CoT checkpoint is already relatively strong. Finally, we observe no statistically significant difference between the CoT (RS-SFT+RS-DPO) and the standard (SFT+RS-DPO) variant, suggesting that CoT does not provide a meaningful additional benefit in this setting.

Table 22 shows the statistical significance of comparing the base untrained model Qwen3-4B with its METALINT variants (SFT and SFT+RS-DPO), and the Qwen3-4B CoT model with METALINT w/ CoT (RS-SFT and RS-SFT+RS-DPO). The SFT variant yields significant gains in detection and localization recall, but not in localization precision. The SFT+RS-DPO model improves significantly across all three metrics. In contrast, training RS-SFT from the Qwen3-4B w/ CoT base does not yield significant improvements. However, the RS-SFT+RS-DPO variant produces significant gains in localization precision and recall, but not detection. These results suggest that while SFT

alone offers limited generalization, combining it with DPO reliably improves localization and can significantly boost detection when starting from a weaker base model.

Table 23 shows the statistical significance results when comparing the METALINT (SFT+RS-DPO) and METALINT w CoT (RS-SFT+RS-DPO) variants against various baselines. Here we want to highlight that METALINT offers comparable performance across two out of three or all three metrics against several 32B models that outperform it like Qwen3-32B, Qwen3-32B w CoT, Qwen2.5Coder-32B and R1-Distill-Qwen-32B. Also the METALINT non CoT (SFT+RS-DPO) variant has no significant difference in performance compared to o3-mini, solidifying that **METALINT without CoT has generalized to the point of being as capable as o3-mini** (even though the Qwen3-4B models without CoT and Qwen3-4B model with CoT perform worse than it with the difference being statistically singificant in Table 20).

Table 24 shows the effect of using a CoT for the Qwen3 model families and we notice that using a CoT leads to singificant gains for all metrics for the 4B and 8B models indicating that for smaller models CoTs might be essential for good performance on this task. However the 14B and 32B model only show statistically significant improvement in localization precision with the CoT indicating that the CoT might offer limited benefit for larger models.

Table 25 shows the effect of varying model scale for the Qwen3, Qwen2.5, Qwen2.5Coder, and DeepSeek-R1-Distill-Qwen families. For Qwen3 we see benefits moving from 4B to 8B abd 8B to 14B but no statistically significant difference moving from 14B to 32B when not using a CoT. Wehn using a CoT for Qwen3 we notice that the performance differences are rarely different in terms of statistical significant except for localizaiton performance between 4B and 8B and 8B and 14B. For R1-Distill-Qwen family we notice a significant difference moving from 14B to 32B but not for 7B to 14B. For the Qwen2.5Coder family we notice difference across all model scales, but the trend is weird with a big drop in performance from 3B to 7B and then a slow climb back to great performance around 32B. We notice that for the Qwen2.5 family which shows relatively reasonable trends with model scale, the performance differences are statistically singificant execpt for the performance gain from 14B to 32B being significant only for recall. To conclude the trends across model scales vary a lot across model families but in general the model size does help but differences may be smaller if the models are capable of reasoning and use a CoT.

Table 26 shows comparison between the GPT models. We only compared GPT-4o and its successor GPT-4.1 and o3-mini against o4-mini and the results show that GPT-4.1 is only significantly better for localization recall while o4-mini is beter than o3-mini for overall localization but not for detection.

### E.3    FAILURE ANALYSIS OF METALINT COT MODEL VS NON COT MODEL

We observe that a significant portion of the lower detection recall of the CoT METALINT Qwen3-4B model, relative to its non CoT counterpart, can be attributed to its higher tendency to predict NO VIOLATIONS FOUND in cases that do, in fact, contain violations. Specifically, the CoT model fails to flag violations in 89 additional instances compared to the non CoT model, amounting to nearly 17% of the evaluation set (89 out of 536 examples).

The idiom wise distribution of these missed violations is shown in Figure 4. While the failure distribution follows a somewhat long tail pattern, the most significant drops occur for PEP 614, PEP 616, and PEP 593. Notably, if the CoT model matched the non CoT model's performance on just these three PEPs, its detection recall would rise to 0.605, surpassing that of all open source baselines evaluated.

Upon inspecting CoT traces for these and other idioms (see examples in Table 27), we identify several recurring failure modes: 1) Ambiguity in interpreting the idiom specification. For example, in PEP 614, which targets decorators with complex expressions, the CoT model often labels expressions that humans consider complex as simple. 2) Overthinking and repetitive reasoning traces, particularly for PEP 616. 3) Skipping or entirely missing lines that contain violations, again observed in PEP 616. 4) Underspecified idioms. For instance, in PEP 593, which recommends using the `Annotated` type from the `typing` module to attach metadata to type hints, the spec lacks clarity and concrete examples, making it hard to learn what constitutes a violation.

We also find similar issues in idioms like PEP 487, which discourages the use of metaclasses for simple customization tasks that could be handled via `__init_subclass__` or `__set_name__`. The CoT model often misclassifies such "simple" use cases as complex.

Overall, these patterns suggest that the CoT model applies the idiom specifications more conservatively, resulting in higher precision but at the cost of reduced recall.

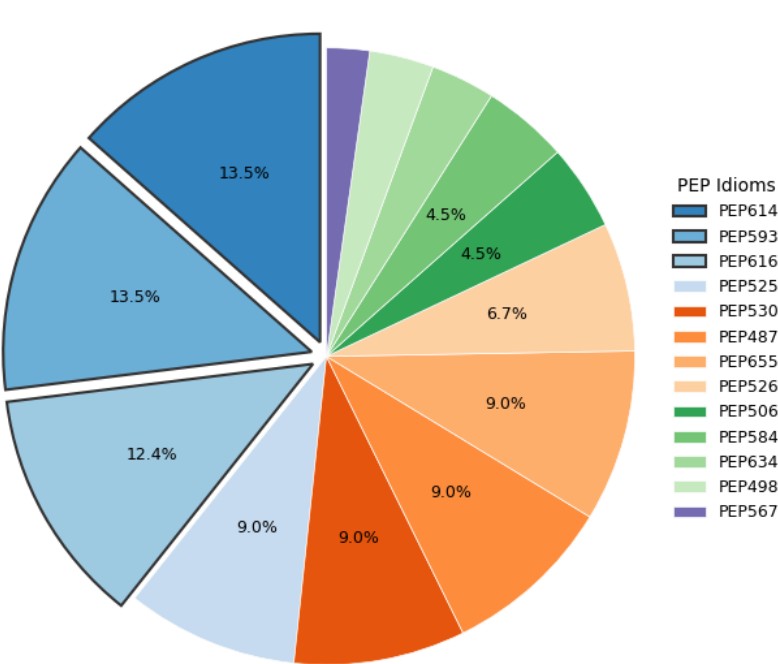

Figure 4: Distribution of comparative failures of the CoT METALINT Qwen3-4B model relative to its non-CoT variant. While errors span a long tail across many PEPs, the majority are concentrated in three: PEP614, PEP593, and PEP616, which motivates our focused analysis on these cases.

| JEP# | JEP Title | Definition | Example(s) | Tree Sitter Queries |
|---|---|---|---|---|
| 394 | PatternMatching InstanceOf (Before) | Usage of the old pattern of testing with instanceof followed by a manual cast to extract and operate on the object. This pattern is verbose and repetitive. Flag the instanceof expression check within a conditional statement and the accompanying cast expression in the body of the conditional statement. | public class ShapeExample { static double getPerimeter(Object obj) { if (obj instanceof Rectangle) { Rectangle r = (Rectangle) obj; return 2 * r.length() + 2 * r.width(); } else if (obj instanceof Circle) { Circle c = (Circle) obj; return 2 * c.radius() * Math.PI; } else { throw new IllegalArgumentException( "Unrecognized shape"); } } } | (if_statement
condition: (parenthesized_expression
(instanceof_expression
left: (identifier) @H1
right: (type_identifier) @H2
)
) @jep_394_before_instanceof_expression.part1
consequence: (block
((local_variable_declaration
type: (type_identifier) @H3
declarator: (variable_declarator
value: (cast_expression
type: (type_identifier) @H4
value: (identifier) @H5
)
)
)(#eq? @H1 @H5) (#eq? @H2 @H3) (
#eq? @H3 @H4)
) @jep_394_before_instanceof_expression.part2
)
) |
| 394 | PatternMatching InstanceOf (After) | Replaces verbose instanceof tests plus manual casting into a concise form that tests and declares a typed variable in one step, for example, "if (obj instanceof String s)" which improves readability, reduces boilerplate, and introduces flow-scoped pattern variables. Flag only the line containing the combined instancesof test and casting within the conditional statement. | public class ShapeExample { static double getPerimeter(Object obj) { if (obj instanceof Rectangle r) { return 2 * r.length() + 2 * r.width(); } else if (obj instanceof Circle c) { return 2 * c.radius() * Math.PI; } else { throw new IllegalArgumentException( "Unrecognized shape"); } } } | [
(instanceof_expression
left: (_)
right: (type_identifier)
name: (identifier)
) @jep_394_after_instanceof_expression
] |
| 378 | TextBlocks (Before) | Multiline strings represented using concatenated string literals, requiring explicit newline escape sequences (\n) and manual concatenation with the + operator. This approach is verbose and error-prone. Flag cases where a variable declaration or method invocation uses concatenated string literals instead of multiline strings. | String html = "\<html\>\n" + " \<body\>\n" + " \<p\>Hello, world!\</p\>\n" + " \</body\>\n" + "\</html\>\n"; | [
(local_variable_declaration
declarator: (variable_declarator
name: (identifier)
value: [
(binary_expression
...
)
] @jep_378_before_concatenated_string_literals
] |
| 378 | TextBlocks (After) | Use of multiline string literal enclosed by triple double-quote marks ("""), allowing for cleaner and more readable representation of multiline strings without explicit escape sequences. Flag cases that use triple double-quote marks for multiline strings in variable declarations or method invocations. | String html = """ \<html\> \<body\> \<p\>Hello, world!\</p\> \</body\> \</html\> """; | [
(string_literal) @jep_378_after_text_block
(#match?
@jep_378_after_text_block "^\"\"\"")
] |
| 361 | Switch Expressions (Before) | Misuse of switch statement with fall-through behavior 'for pattern matching. This pattern is verbose and error prone. You should flag case statements with empty bodies that are misusing fall-through behavior. | int numLetters; switch (day) { case MONDAY: case FRIDAY: case SUNDAY: numLetters = 6; break; case TUESDAY: numLetters = 7; break; ... throw new IllegalStateException( "Unexpected value: " + day); } | jep_361_before_custom_detectors |

Table 5: **JEP Idiom Specifications (1/3):** This table presents 15 idioms across 8 JEPs, including both "before" (old best practice) and "after" (updated best practice) patterns. The JEP# column lists the JEP number, the JEP title specifies the idiom topic, and the parenthesized value indicates whether it is a before or after pattern. The Definition, Example, and Tree-Sitter Queries columns provide the idiom definition, minimal Java examples shown to the LLM as instructions, and the queries used to flag idioms for synthetic data creation.

| JEP# | JEP Title | Definition | Example(s) | Tree Sitter Queries |
|------|-----------|------------|------------|---------------------|
| 361 | Switch Expressions (After) | Use of switch expressions, allowing a return value. Employs the -> syntax for case labels, eliminating fall-through behavior. Flag statements that use the arrow operator "->" or "yield" syntax. | Example 1:

int numLetters = switch (day) {
case MONDAY, FRIDAY, SUNDAY ->6;
case TUESDAY ->7;
case THURSDAY, SATURDAY ->8;
case WEDNESDAY ->9;
...
Example 7:

String category = switch (age) {
case 0, 1, 2, 3, 4, 5 ->"Toddler";
case 6, 7, 8, 9, 10, 11, 12 ->"Child";
case 13, 14, 15, 16, 17, 18, 19 ->"Teenager";
default ->"Adult";
};

Example 8:

String response = switch (input) {
case "yes" ->"Affirmative";
case "no" ->"Negative";
default ->"Unrecognized input";
}; | [
(yield_statement
) @jep_361_after_yield
(switch_rule
(switch_label)
"->" @jep_361_after_arrow
)
(switch_rule
(switch_label)
"->" ;; ensures it's not arrow
(block (yield_statement
) @jep_361_after_yield)
)
] |
| 314 | UnicodeLang TagExtensions (After) | Use java.util.Locale with additional BCP 47 Unicode extensions (cu, fw, rg, tz) in Java 10 to customize locale behavior like currency (java.util.Currency), first-day-of-week (java.time.temporal.WeekFields), region override (java.text.NumberFormat.getInstance), and time zone (java.time.format.DateTimeFormatter). Flag imports and function calls related to these. | Example 1: Currency Type (cu)

import java.util.Locale;
import java.util.Currency;

public class Foo {
void bar() {
Locale locale = Locale.forLanguageTag(
"en-US-u-cu-EUR");
Currency c = Currency.getInstance(locale);
System.out.println(c);
}
}
...
Example 4: Time Zone (tz)

import java.util.Locale;
import java.time.format.DateTimeFormatter;
import java.time.ZonedDateTime;

public class Foo {
void bar() {
Locale locale = Locale.forLanguageTag(
"en-US-u-tz-Asia-Tokyo");
DateTimeFormatter fmt =
DateTimeFormatter.ofPattern(
"yyyy-MM-dd HH:mm z").withLocale(locale);
System.out.println(
fmt.format(ZonedDateTime.now()));
}
} | [
(import_declaration
((scoped_identifier
scope: (scoped_identifier
) @H2
name: (identifier) @H1
) (#eq? @H2 "java.util") (
#eq? @H1 "Currency"))
) @jep_314_after_currency_import

((method_invocation
object: (identifier) @H7
name: (identifier) @H8
) (#eq? @H7 "Currency") (
#eq? @H8 "getInstance")
) @jep_314_after_currency...

...
((method_invocation
object: (identifier) @H13
name: (identifier) @H14
) (
#eq? @H13 "NumberFormat") (
#eq? @H14 "getInstance")
) @jep_314_after_number_format...
] |
| 395 | RecordClass (Before) | Use of simple data aggregates with traditional classes which could be replaced with a record class. This approach requires explicit declarations of fields, constructors, and accessor methods, leading to verbose and repetitive code. Flag non record classes containing equals(), hashCode(), and toString() methods. | public class Point {
private final int x;
private final int y;
...

public int x() {
return x;
}
...
@Override
public String toString() {
return "Point{x=" + x + ", y=" + y + "}";
}

@Override
public boolean equals(Object obj) {
...
}

@Override
public int hashCode() {
return Objects.hash(x, y);
}

} | [
(class_declaration
body: (class_body
(constructor_declaration
) @H1
(method_declaration
name: (identifier) @H2
)
) (#match? @H2 "^(
hashCode—equals—toString)$")
) @jep_395_before_record_like_class
] |
| 395 | RecordClass (After) | Use of record class. Record classes introduce a concise syntax for defining immutable data aggregates, automatically generating canonical constructors, accessors, equals(), hashCode(), and toString() methods, thereby reducing boilerplate code and enhancing readability. | Example 1 (Record Declaration):

record Point(int x, int y) {}

Example 2 (Record Declaration):

record Rectangle(double length, double width) {}
... | [
(record_declaration
) @jep_395_after_record_...
] |

Table 6: **JEP Idiom Specifications (2/3)**

| JEP# | JEP Title | Definition | Example(s) | Tree Sitter Queries |
|---|---|---|---|---|
| 409 | Sealed Class (Before) | Use of abstract classes with private constructors to simulate sealed classes using package-private visibility to restrict subclassing. This approach lacks explicit language support and is error-prone. Switch to sealed classes. Flag abstract classes with private constructors. | public abstract class Shape {
private Shape() {}
}
public class Circle extends Shape {
/* Implementation */
}
public class Square extends Shape {
/* Implementation */
} | [
((class_declaration
(modifiers) @H1
name: (identifier) @H4
body: (class_body
(constructor_declaration
(modifiers) @H2
name: (identifier) @H3
... )(#match? @H1 "abstract")
(#eq? @H3 @H4)
(#match? @H2 "private")
) @jep_409_before_abstract_class...
] |
| 409 | Sealed Class (After) | Use of sealed classes to explicitly define which classes or interfaces can extend or implement them using the sealed modifier and the permits clause. This feature enhances type safety and exhaustiveness checking. Flag class declarations with the sealed or non-sealed modifiers and lines with the permit clause. | public sealed class Shape
permits Circle, Square {
/* Implementation */
}
public final class Circle extends Shape {
/* Implementation */
}
public final class Square extends Shape {
/* Implementation */
} | [
(permits
) @jep_409_after_permits_clause
((class_declaration
(modifiers) @H1
)(#match? @H1 "sealed")
) @jep_409_after_sealed_modifier
] |
| 406 | Pattern Matching Switch (Before) | Use of a sequence of if-else if statements to test an object's type via instanceof, with a manual cast, to handle each case separately. This approach is verbose, error-prone, and lacks exhaustiveness checking or compiler assistance for missing cases. Flag if or else-if statements that contain instanceof statements with a manual cast in the statement body. | static String formatter(Object o) {
if (o instanceof Integer) {
Integer i = (Integer) o;
return String.format("int %d", i);
} else if (o instanceof Long) {
Long l = (Long) o;
return String.format("long %d", l);
} else if (o instanceof String) {
String s = (String) o;
return String.format("String %s", s);
} else {
return o.toString();
}
} | (if_statement
condition: (parenthesized_expression
(instanceof_expression
left: (identifier) @H1
right: (type_identifier) @H2
...
) @jep_406_before_if_else_if_....
consequence: (block
((local_variable_declaration
type: (type_identifier) @H3
declarator: (variable_declarator
value: (cast_expression
type: (type_identifier) @H4
value: (identifier) @H5
...
(#eq? @H1 @H5)
(#eq? @H2 @H3)
(#eq? @H3 @H4)
) @jep_406_before_if_else_if...
)
) |
| 406 | Pattern Matching Switch (After) | Use of a switch expression or statement with case labels containing type patterns (and optionally a guard), binding the matched variable within the branch. This style is more concise, expressive, and opens opportunities for compiler-checked exhaustiveness and performance optimizations. Flag switch labels (case statements) with patterns, null literals or paranthesized expressions but skip default switch labels/cases. | Example 1:
static String formatter(Object o) {
return switch (o) {
case Integer i ->String.format("int %d", i);
case Long l ->String.format("long %d", l);
case String s ->String.format("String %s", s);
default ->o.toString();
};
}
Example 2:
static String checkShape(Object o) {
return switch (o) {
... | [
(switch_label
(null_literal)
) @jep_406_after_null_case
(switch_label
(pattern)
) @jep_406_after_switch_pattern
(switch_label
(parenthesized_expression)
) @jep_406_after_paranthesized_pattern
(switch_label
(binary_expression)
) @jep_406_after_binary_expression
] |
| 323 | LocalVar Syntax Lambda Params (Before) | Use of implicitly typed lambda expressions with omitted type declarations. These lambda expressions rely solely on parameter names. This approach prioritizes brevity but lacks explicit type information. Flag full lambda expressions without type declarations. | Example 1:
xs.stream().filter((a, b) ->a <b).forEach(
System.out::println);
...
Example 4:
xs.stream().filter((a) ->a >10).forEach(
System.out::println); | jep_323_before_custom_detector |
| 323 | LocalVar Syntax Lambda Params (After) | Use of explicit type declarations for lambda parameters, enhancing code clarity and enabling better static analysis tools. Flag full lambda expressions with explicit type declarations using formal parameters (var). | Example 1:
xs.stream().filter(
(var a, var b) ->a.compareTo(b) <0).forEach(
System.out::println);
...
Example 4:
xs.stream().filter((var a) ->a >10).forEach(
System.out::println); | (lambda_expression
parameters: (formal_parameters
(formal_parameter
type: (type_identifier) @H1
(#eq? @H1 "var"))
)) @jep_323_after_local_var_lambda |

Table 7: **JEP Idiom Specifications (3/3)**

| JEP # | Before | After | Title | JDK# | Release Date |
|-------|--------|-------|-------|------|--------------|
| 409 | Yes | Yes | Sealed Classes | 17 | 14 Sept 2021 |
| 406 | Yes | Yes | Pattern Matching for switch | 17 | |
| 395 | Yes | Yes | Records | 16 | 16 Mar 2021 |
| 394 | Yes | Yes | Pattern Matching for instanceof | 16 | |
| 378 | Yes | Yes | Text Blocks | 15 | 15 Sept 2020 |
| 361 | Yes | Yes | Switch Expressions | 14 | 17 Mar 2020 |
| 323 | Yes | Yes | Local-Variable Syntax for Lambda Parameters | 11 | 25 Sept 2018 |
| 314 | No | Yes | Additional Unicode Language-Tag Extensions | 10 | 20 Mar 2018 |

Table 8: List of JEPs addressed by our tree-sitter synthetic data. The JEP# and Title column indicate the number and title of the JEP while JDK# and Release Date indicate the JDK needed for compilation to be able to use the JEP features. The Before and After columns indicate whether we include rules/patterns to flag the old idiom or new idiom introduced by the JEP.

| Training Set Idioms | Test Set Idioms |
|---------------------|-----------------|
| **PyFlakes:** 
 F405, F501, F502, F601, F621 | **PyFlakes:** 
 F403, F406, F503, F602, F622 |
| **pycodestyle:** 
 E402, E701, E721, E741, E743 | **pycodestyle:** 
 E401, E702, E722, E731, E742 |
| **Naming:** 
 N801, N802, N803, N804, N805, 
 N806, N807, N811, N812, N813 | **Miscellaneous:** 
 ERA001, C901, I001, I002, BLE001 
 (shared with training) |
| **pyupgrade:** 
 UP001, UP002, UP003, UP004, UP005, UP006, 
 UP007, UP008, UP009, UP010, UP011, 
 UP040, UP044, UP045, UP046, UP047 | **flake8 annotations:** 
 ANN001, ANN002, ANN003, ANN201, ANN202, 
 ANN204, ANN205, ANN206 |
| **Miscellaneous:** 
 ERA001, C901, I001, I002, BLE001 | **flake8 async:** 
 ASYNC100, ASYNC105, ASYNC109, ASYNC110, 
 ASYNC115, ASYNC116, ASYNC210, ASYNC220, 
 ASYNC221, ASYNC222, ASYNC230, ASYNC251 |
| **Bugbear:** 
 B002, B003, B004, B005, B006, 
 B007, B008, B009, B010, B012 | **flake8 bandit:** 
 S102, S103, S104, S105, S106, 
 S107, S108, S110, S112, S113, 
 S201, S202, S301, S302, S303 |

Table 9: Ruff idioms included in the supervised training and transfer evaluation test sets. Test set idioms span both overlapping linters and novel ones not seen during training.

| Fraction of NV data | Detection | | | Localization | | |
|---|---|---|---|---|---|---|
| | $P_{Det}$ | $R_{Det}$ | $F_{Det}$ | $P_{Loc}$ | $R_{Loc}$ | $F_{Loc}$ |
| 0% | 0.6268 | 0.9577 | 0.7577 | 0.6777 | 0.6932 | 0.6854 |
| 2% | 0.671 | 0.9128 | 0.7734 | 0.6681 | 0.6812 | 0.6746 |
| 5% | 0.7469 | 0.8315 | 0.7869 | 0.6527 | 0.6696 | 0.6611 |
| 10% | 0.7584 | 0.8114 | 0.784 | 0.6263 | 0.6474 | 0.6367 |
| 20% | 0.8382 | 0.7227 | 0.7762 | 0.5721 | 0.5815 | 0.5768 |
| 40% | 0.8683 | 0.5618 | 0.6822 | 0.4683 | 0.4735 | 0.4709 |
| 100% | 0.8565 | 0.4152 | 0.5593 | 0.4041 | 0.4056 | 0.4048 |

Table 10: Effect of varying the fraction of NO VIOLATIONS FOUND instances in the training data for METALINT Qwen3-4B model without CoT. Including 0% yields the highest recall and best line-level localization but reduces precision due to more false positives and lower accuracy in predicting NO VIOLATIONS FOUND. Conversely, including 100% improves precision but leads to reduced recall and localization performance. All rows report the performance at the best training step, selected based on a balance of detection and localization F-score on the Ruff Idiom Transfer test set.

| Fraction of NV data | Detection | | | Localization | | |
|---|---|---|---|---|---|---|
| | $P_{Det}$ | $R_{Det}$ | $F_{Det}$ | $P_{Loc}$ | $R_{Loc}$ | $F_{Loc}$ |
| 2% | 0.9226 | 0.8901 | 0.906 | 0.7688 | 0.7638 | 0.7663 |
| 5% | 0.9234 | 0.8643 | 0.8929 | 0.771 | 0.7571 | 0.764 |

Table 11: Effect of varying the fraction of NO VIOLATIONS FOUND instances in the training data for METALINT Qwen3-4B model with CoT. We perform limited ablations because of the insights from the non CoT model training.

| Fraction of NV data | Detection | | | Localization | | |
|---|---|---|---|---|---|---|
| | $P_{Det}$ | $R_{Det}$ | $F_{Det}$ | $P_{Loc}$ | $R_{Loc}$ | $F_{Loc}$ |
| 1% | 0.654 | 0.6468 | 0.6504 | 0.491 | 0.4788 | 0.4848 |
| 2% | 0.6636 | 0.6057 | 0.6333 | 0.4869 | 0.4745 | 0.4806 |

Table 12: Effect of varying the fraction of NO VIOLATIONS FOUND instances in the training data for METALINT Llama3.2-3B-Instruct model. We perform limited ablations because of the insights from the non CoT model training.

| PEP | Description | Heuristics | Example |
|---|---|---|---|
| 506 | Adds secrets module to the standard library for cryptographically secure random value generation | Conjunction of 2 conditions:
1. Presence of "random" module imports
2. Presence of "random" function usage | characters = string.ascii_letters + string.punctuation + string.digits password = "".join(random.choice (characters) for x in range(16))
Use instead:
characters = string.ascii_letters + string.punctuation + string.digits password = "".join(secrets.choice (characters) for x in range(16)) |
| 557 | Introduces the dataclasses module, enabling automatic generation of common boilerplate methods for classes | Conjunction of 2 conditions:
1. There is a class with manual implementation of "__init__" method
2. On the same class there is manual implementation of common special methods or comparison methods that follow standard data storage patterns. | "class Point:
def __init__(self, x, y):
self.x = x
self.y = y
def __repr__(self):
return f"Point(x={self.x}, y={self.y})""
Use instead:
from dataclasses import dataclass
@dataclass
class Point:
x: int
y: int" |
| 655 | Introduces Required[] and NotRequired[] type qualifiers to replaces cumbersome TypedDict inheritance patterns. | Conjuction of:
1. "TypedDict" defined with inheritance pattern.
2. total=False parameter usage in class definition | class _MovieBase(TypedDict): # implicitly total=True
title: str
class Movie(_MovieBase, total=False):
year: int
Use instead:
class Movie(TypedDict):
title: str
year: NotRequired[int] |
| 634 | Introduced structural pattern matching, enabling more expressive and concise ways to match data structures and control flow. | Multiple consecutive if-elif-else statements that compare a single variable against different values with dysjunction of 2 conditions:
1. Length of ladder (number of conditons at the "top level" + one level in) >= 6
2. Depth of ladder (degree of nesting) >=3 | "def handle_response(response):
if isinstance(response, dict):
if ""error"" in response:
print(f"Error: {response['error']}"")
elif ""data"" in response:
print(f"Data: {response['data']}"")
else:
print(""Unknown response format"")
elif isinstance(response, list):
print(""List of items:"", response)
else:
print(""Invalid response type"")
Use instead:
def handle_response(response):
match response:
case {""error"": error_message}:
print(f"Error: {error_message}"")
case {""data"": data_content}:
print(f"Data: {data_content}"")
case list(items):
print(""List of items:"", items)
case _: print(""Invalid response type"")" |
| 614 | Removes previous restrictions on decorator syntax. Before, only simple names or dotted names were valid decorators. After 614, any valid expression can be used as a decorator | Conjunction of 2 conditions:
1. A decorator is applied using a name (e.g., @decorator) where that name is assigned earlier in the code.
2. The assignment value is an expression of type Call, Attribute, or Subscript (e.g., deco = factory(), deco = module.decorator, deco = decorators[i]). | # def uppercase(func):
def wrapper(*args, **kwargs):
return func(*args, **kwargs).upper()
return wrapper
@uppercase
def greet():
return "hello"
Use Instead:
deco = [uppercase]
@deco[0]
def greet2():
return "hi" |
| 616 | Replaces manual slicing with dedicated methods | dysjunction of 2 conditions:
1. There is a "check" with startswith or endswith on a given variable x.
2. On the same variable x check if there is an "edit" using a program slicing syntax or using "replace()". | if s.startswith(prefix): s = s[len(prefix):]
Use instead:
s = s.removeprefix(prefix)
OR
s[:-len(suffix)]
Use instead:
s.removesuffix(suffix) |
| 584 | Introduces the binary operators — (merge) and —= (update) on dict (and other built-in mapping types), providing an expressive, in-place-or-new-object way to combine dictionaries. | disjunction of two conditions:
1. A copy-and-update sequence on the same variable or in close proximity: d = d1.copy() followed by d.update(d2)
2. A dictionary literal using multiple unpackings {**d1, **d2}, indicating ad-hoc merging rather than the new operators | d1 = {'a': 1, 'b': 2} d2 = {'c': 3, 'd': 4}
merged = d1.copy()
merged.update(d2)
d1 = {'a': 1, 'b': 2} d2 = {'c': 3, 'd': 4}
merged = {**d1, **d2}
d1 = {'a': 1, 'b': 2} d2 = {'c': 3, 'd': 4}
merged = dict(list(d1.items()) + list(d2.items()))
Use instead:
d1 = {'a': 1, 'b': 2} d2 = {'c': 3, 'd': 4}
merged = d1 — d2
d1 = {'a': 1, 'b': 2} d2 = {'c': 3, 'd': 4}
d1 —= d2 # d1 is now {'a': 1, 'b': 2, 'c': 3, 'd': 4} |

Table 13: High recall heuristics used to find instances of PEP violations that human annotators vet

| PEP | Description | Heuristics | Example |
|---|---|---|---|
| 570 | Introduces new syntax (the / marker) in Python function signatures to specify positional-only parameters, ensuring that certain arguments can only be supplied by their position and not as keywords | Conjunction of the following conditions: 1. Have only positional-or-keyword parameters (without *args, **kwargs, keyword -only parameters, or the '/' marker), 2. Include 2 to 4 parameters, all of which have no default values | def compute_area(width, height): return width * height area = compute_area(width=5, height=10) print("Area:", area) Use instead: def compute_area(width, height, /): return width * height area = compute_area(5, 10) print("Area:", area) |
| 567 | Adds the contextvars module, enabling context-local variables for managing dynamic state. | Dysjunction of the following conditions: 1. Look for import threading together with threading.local() object creation and use. 2. Find global statements or assignment to variables at the module level that are accessed or mutated in functions, especially as shared state. 3. Identify async functions or classes where context or state variables are passed as parameters (e.g., def func(context, ...) or async def func(context, ...)), not as context-local variables. | import threading _thread_local = threading.local() def set_context(value): _thread_local.value = value def get_context(): return getattr(_thread_local, 'value', None) Use instead: from contextvars import ContextVar context_var = ContextVar('value') def set_context(value): context_var.set(value) def get_context(): return context_var.get() |
| 530 | Enables the use of "async for" and "await" in list, set, and dict comprehensions as well as in generator expressions, providing concise asynchronous data processing within comprehensions | Dysjunction of the following conditions: 1. "async" def functions that uses "async for" loops to build lists, sets, or dicts. 2. "async for" loops, followed by methods like result.append(...), result.extend(...), or result[key] = .... 3. Comprehensions written without the "async for" clause despite being inside an "async def" | result = [] async for i in aiter(): if i % 2: result.append(i) Use instead: result = [i async for i in aiter() if i % 2] |
| 525 | Introduces the ability to define asynchronous generator functions using the async def and yield syntax, enabling concise, native support for asynchronous iteration. | Dysjunction of the following conditions: 1. classes defining both "__aiter__" and "__anext__" methods, especially where the class is used solely to produce a sequence of values asynchronously. 2. async def functions that create and return custom iterator classes instead of using async def with yield. | class Ticker: """Yield numbers from 0 to 'to' every 'delay' seconds.""" def __init__(self, delay, to): self.delay = delay self.i = 0 self.to = to def __aiter__(self): return self async def __anext__(self): i = self.i if i >= self.to: raise StopAsyncIteration self.i += 1 if i: await asyncio.sleep(self.delay) return i Use instead: async def ticker(delay, to): """Yield numbers from 0 to 'to' every 'delay' seconds.""" for i in range(to): yield i await asyncio.sleep(delay) |
| 520 | Ensures that the order in which attributes are defined within a class body is preserved in the resulting class object, making the attribute order predictable and consistent. | Dysjunction of the following conditions: 1. Uses sorted() or otherwise processes class.__dict__.keys() to impose attribute order. 2. Attribute names are tracked in a list or similar structure solely to maintain definition order. 3. Custom metaclass logic or "__prepare__" implementations created to preserve the order of class attributes. | class Person: name = "Alice" age = 30 city = "Wonderland" def display_attributes(self): # Manually sorting keys for key in sorted(self.__class__.__dict__.keys()): if not key.startswith("__"): print(key, getattr(self, key)) Use instead: class Person: name = "Alice" age = 30 city = "Wonderland" def display_attributes(self): # Directly iterate over the preserved definition order for key in self.__class__.__definition_order__: print(key, getattr(self, key)) |
| 498 | Introduces f-strings (formatted string literals) as a new, concise, and efficient way to embed Python expressions inside string literals using the f" prefix. | Dysjunction of the following conditions: 1. Occurrences of string literals with .format(...) applied, especially where keys or variables match braces in the string 2. String literals concatenated using "+" with variables. 3. Uses of the "%" operator for string formatting, | name = "Alice" age = 30 greeting = "Hello, " + name + "! You are " + str(age) + " years old." Use instead: name = "Alice" age = 30 greeting = f"Hello, {name}! You are {age} years old." OR value = 12.3456 formatted = "The value is {:.2f}".format(value) Use instead: value = 12.3456 formatted = f"The value is {value:.2f}" |

Table 14: High recall heuristics used to find instances of PEP violations that human annotators vet

| PEP | Description | Heuristics | Example |
|---|---|---|---|
| 487 | Makes customizing class creation and subclass initialization easier by introducing __init_subclass__ and __set_name__, eliminating the need for most custom metaclasses | Disjunction of the following conditions: 1. Custom metaclasses defined to execute code during class creation or subclassing (e.g., overriding __new__, __init__, or __call__ in metaclasses) instead of using __init_subclass__. 2. Descriptor classes lacking __set_name__ method and employing manual workarounds to determine their assigned attribute names. 3. Classes or frameworks manually tracking or registering subclasses via metaclass hooks instead of leveraging __init_subclass__. | class Meta(type): def __new__(meta, name, bases, namespace): for key, value in namespace.items(): if isinstance(value, Descriptor): value.name = key return super().__new__ (meta, name, bases, namespace) class MyClass(metaclass=Meta): attr = Descriptor() Use instead: class Descriptor: def __set_name__(self, owner, name): self.name = name class MyClass: attr = Descriptor() OR class PluginBase(type): plugins = {} def __new__(meta, name, bases, namespace): if name != 'Plugin': meta.plugins[name] = namespace['priority'] return super().__new__ (meta, name, bases, namespace) class Plugin(metaclass=PluginBase): priority = 0 class HighPriority(Plugin): priority = 10 Use instead: class Plugin: plugins = {} priority = 0 def __init_subclass__(cls, **kwargs): super().__init_subclass__(**kwargs) cls.plugins[cls.__name__] = cls.priority class HighPriority(Plugin): priority = 10 |
| 593 | Introduces flexible function and variable annotations via typing.Annotated, which lets you attach context-specific metadata to type hints (e.g., validation constraints, units) | Conjunction of 2 conditions: 1. Type hints are already present in function arguments, return types, or variable annotations. 2. Nearby comments/docstrings (within ±2 lines) contain metadata-like patterns such as "min", "max", "nullable", "regex", "enum", "unit", "deprecated", etc. | # max 100, min 1 def set_age(age: int) ->None: pass  Use instead:  from typing import Annotated Age = Annotated[int, "min=1", "max=100"] def set_age(age: Age) ->None: pass |
| 526 | introduces explicit variable annotations, allowing type hints directly on variable declarations for local, global, and class variables in Python | Disjunction of the following conditions: 1. Variables assigned values with a type comment (e.g., x = 0 # type: int) instead of using annotation syntax. 2. Identify variable assignments, especially class and instance attributes, that lack any type annotation (e.g., name = "" in class bodies). 3.Module-level variables assigned values without accompanying type hints— especially in type-annotated codebases. | # type: List[int] numbers = [] Use instead: numbers: List[int] = [] OR class Player: # type: str
name = "Guest" Use instead: class Player: name: str = "Guest" |
| 589 | Introduces TypedDict, enabling precise type hints for dictionaries with a fixed set of string keys, improving static type checking and readability in Python code. | Disjunction of the following conditions: 1. Dictionary literals or variables consistently using the same fixed set of string keys without accompanying TypedDict annotations. 2. Functions annotated with broad dictionary types like Dict[str, Any], dict, or untyped parameters/returns that actually expect dictionaries with a known fixed set of keys. 3. Explicit key presence checks or accessing dictionary keys repeatedly that suggest a structured dictionary shape. | movie = {'name': 'Blade Runner', 'year': 1982} Use instead: from typing import TypedDict class Movie(TypedDict): name: str year: int movie: Movie = {'name': 'Blade Runner', 'year': 1982} |
| 572 | Introduces the assignment expression operator := (the "walrus operator"), allowing assignment to variables within expressions, | 1. Patterns where a value is first assigned to a variable, and then immediately checked or used in the next line or inside a loop, list comprehension, or condition. 2. separate assignment and conditional test statements | match = pattern.search(data) if match is not None: process(match) Use instead: if (match := pattern.search(data)) is not None: process(match) |

Table 15: High recall heuristics used to find instances of PEP violations that human annotators vet

| Model | In-Domain | | | Near Transfer | | | Far Transfer | | |
|---|---|---|---|---|---|---|---|---|---|
| | $P_{Det}$ | $R_{Det}$ | $F_{Det}$ | $P_{Det}$ | $R_{Det}$ | $F_{Det}$ | $P_{Det}$ | $R_{Det}$ | $F_{Det}$ |
| Qwen3-4B | 0.45 | 0.14 | 0.22 | 0.58 | 0.24 | 0.34 | 0.54 | 0.29 | 0.38 |
| +SFT | **0.93 (+0.48)** | 0.74 (+0.6) | 0.83 (+0.61) | **0.89 (+0.31)** | 0.24 (+0) | 0.38 (+0.04) | 0.72 (+0.18) | 0.27 (-0.02) | 0.39 (+0.01) |
| +RS-DPO | 0.72 (+0.27) | 1 (+0.86) | **0.83 (+0.61)** | 0.76 (+0.18) | **0.8 (+0.56)** | 0.78 (+0.44) | 0.75 (+0.21) | **0.81 (+0.52)** | **0.78 (+0.4)** |
| Qwen3-4B w CoT | 0.87 | 0.5 | 0.63 | 0.95 | 0.88 | 0.91 | 0.87 | 0.68 | 0.76 |
| +RS-SFT | **0.87 (+0)** | 0.73 (+0.23) | 0.8 (+0.17) | **0.97 (+0.02)** | 0.86 (-0.02) | 0.91 (+0) | **0.94 (+0.07)** | 0.82 (+0.14) | 0.88 (+0.12) |
| +RS-DPO | 0.86 (-0.1) | **0.85 (+0.35)** | **0.85 (+0.22)** | **0.97 (+0.02)** | **0.92 (+0.04)** | **0.94 (+0.03)** | 0.92 (+0.05) | **0.86 (+0.18)** | **0.89 (+0.13)** |
| Llama3.2-3B-Instruct | 0.54 | 0.43 | 0.48 | 0.69 | 0.68 | 0.69 | 0.47 | 0.51 | 0.49 |
| +SFT | **0.88 (+0.34)** | **0.87 (+0.44)** | **0.88 (+0.4)** | **0.89 (+0.2)** | 0.44 (-0.24) | 0.59 (-0.1) | 0.61 (+0.14) | 0.27 (-0.24) | 0.37 (-0.12) |
| +RS-DPO | 0.75 (+0.21) | 0.92 (+0.49) | 0.83 (+0.35) | 0.81 (+0.12) | **0.71 (+0.03)** | **0.76 (+0.07)** | **0.61 (+0.14)** | **0.59 (+0.08)** | **0.60 (+0.11)** |

Table 16: **Cross-Idiom Generalization on Python Ruff Idioms by Transfer Setting:** We evaluate the effect of different METALINT training setups (SFT, RS-SFT, and RS-DPO) on Qwen3-4B (with and without reasoning) and Llama3.2-3B. Models are trained on easy synthetic Python Ruff idioms, and the performance is reported on other Ruff idioms with varying levels of transfer - In-Domain, Near Transfer, and Far Transfer (section 4.2).

| Model | Detection | | | Localization | | |
|---|---|---|---|---|---|---|
| | $P_{Det}$ | $R_{Det}$ | $F_{Det}$ | $P_{Loc}$ | $R_{Loc}$ | $F_{Loc}$ |
| Llama3.2-3B-Instruct | 0.7042 | 0.214 | 0.3283 | 0.0691 | 0.0798 | 0.0741 |
| Qwen3-4B | 0.5267 | 0.1715 | 0.2587 | 0.0954 | 0.0824 | 0.0884 |
| Qwen3-4B with CoT | 0.8154 | 0.3986 | 0.5354 | 0.2625 | 0.1467 | 0.1882 |
| Qwen3-8B | 0.8267 | 0.3572 | 0.4988 | 0.1806 | 0.1285 | 0.1501 |
| Qwen3-8B with CoT | 0.8886 | 0.4672 | 0.6124 | 0.3122 | 0.2029 | 0.2459 |
| Qwen3-14B | 0.9021 | 0.4612 | 0.6103 | 0.289 | 0.2521 | 0.2693 |
| Qwen3-14B with CoT | 0.9116 | 0.4857 | 0.6337 | 0.3993 | 0.2915 | 0.3369 |
| Qwen3-32B | 0.9021 | 0.5205 | 0.6601 | 0.2807 | 0.2711 | 0.2758 |
| Qwen3-32B with CoT | 0.9377 | 0.5645 | 0.7048 | 0.4152 | 0.3086 | 0.354 |
| Qwen2.5-3B-Instruct | 0.0667 | 0.0033 | 0.0063 | 0.0036 | 0.0036 | 0.0036 |
| Qwen2.5-7B-Instruct | 0.4333 | 0.1379 | 0.2092 | 0.0585 | 0.0518 | 0.0549 |
| Qwen2.5-14B-Instruct | 0.8017 | 0.4324 | 0.5618 | 0.2389 | 0.2158 | 0.2267 |
| Qwen2.5-32B-Instruct | 0.8667 | 0.2656 | 0.4066 | 0.163 | 0.1477 | 0.155 |
| Qwen2.5Coder-3B-Instruct | 0.7802 | 0.411 | 0.5384 | 0.1257 | 0.0745 | 0.0936 |
| Qwen2.5Coder-7B-Instruct | 0.0667 | 0.0033 | 0.0063 | 0 | 0 | 0 |
| Qwen2.5Coder-14B-Instruct | 0.2 | 0.0443 | 0.0726 | 0.0294 | 0.0264 | 0.0278 |
| Qwen2.5Coder-32B-Instruct | 0.8961 | 0.5328 | 0.6683 | 0.3432 | 0.3077 | 0.3245 |
| DeepSeek-R1-Distill-Qwen-7B with CoT | 0.7143 | 0.2841 | 0.4065 | 0.1064 | 0.1122 | 0.1092 |
| DeepSeek-R1-Distill-Qwen-14B with CoT | 0.69 | 0.2345 | 0.35 | 0.1856 | 0.1245 | 0.149 |
| DeepSeek-R1-Distill-Qwen-32B with CoT | 0.9008 | 0.5899 | 0.713 | 0.4015 | 0.3403 | 0.3684 |
| GPT-oss-20b | 0.8377 | 0.3531 | 0.4968 | 0.251 | 0.1695 | 0.2024 |
| GPT-oss-120b | 0.9157 | 0.6456 | 0.7573 | 0.3991 | 0.3331 | 0.3631 |
| Qwen3-4B METALINT (SFT) (**Ours**) | 0.4333 | 0.0821 | 0.1381 | 0.0432 | 0.0221 | 0.0292 |
| Qwen3-4B METALINT (SFT+RS-DPO) (**Ours**) | 0.7031 | **0.7043** | 0.7037 | 0.3536 | 0.193 | 0.2497 |
| Qwen3-4B METALINT w CoT (RS-SFT) (**Ours**) | 0.7615 | 0.3689 | 0.497 | 0.2785 | 0.1437 | 0.1896 |
| Qwen3-4B METALINT w CoT (RS-SFT+RS-DPO) (**Ours**) | 0.9303 | 0.4958 | 0.6468 | 0.3482 | 0.2169 | 0.2673 |
| Llama3.2-3B-Instruct METALINT (SFT) (**Ours**) | 0.5627 | 0.259 | 0.3547 | 0.1066 | 0.0509 | 0.0689 |
| Llama3.2-3B-Instruct METALINT (SFT+RS-DPO) (**Ours**) | 0.6368 | 0.5614 | 0.5965 | 0.2364 | 0.1263 | 0.1647 |
| o3-mini | 0.8939 | 0.5845 | 0.7068 | 0.3169 | 0.2361 | 0.2706 |
| o4-mini | **0.9667** | 0.5943 | 0.7361 | 0.4131 | 0.3164 | 0.3584 |
| GPT-4o | 0.8938 | 0.6788 | **0.7716** | 0.4461 | 0.332 | 0.3807 |
| GPT-4.1 | 0.907 | 0.646 | 0.7546 | **0.4632** | **0.4673** | **0.4653** |
| GPT-5 (high) | 0.913 | 0.5673 | 0.6998 | 0.4397 | 0.4257 | 0.4326 |

Table 17: Results on the hard PEP benchmark to measure easy to hard generalization.

| Model | Detection | | | Localization | | |
|---|---|---|---|---|---|---|
| | $P_{Det}$ | $R_{Det}$ | $F_{Det}$ | $P_{Loc}$ | $R_{Loc}$ | $F_{Loc}$ |
| Qwen3-4B | 0.538 | 0.2637 | 0.3539 | 0.1396 | 0.1479 | 0.1436 |
| Qwen3-4B + SFT | **0.7686** | 0.3178 | 0.4497 | 0.2976 | 0.296 | 0.2968 |
| Qwen3-4B + SFT + RS-DPO | 0.7469 | **0.8315** | **0.7869** | **0.6527** | **0.6696** | **0.6611** |
| Qwen3-4B w CoT | 0.8812 | 0.6854 | 0.771 | 0.5049 | 0.4878 | 0.4962 |
| Qwen3-4B w CoT + RS-SFT | **0.935** | 0.8183 | 0.8727 | 0.6639 | 0.65 | 0.6569 |
| Qwen3-4B w CoT + RS-SFT + RS-DPO | 0.9234 | **0.8643** | **0.8929** | **0.771** | **0.7571** | **0.764** |
| Llama3.2-3B-Instruct | 0.5092 | 0.5286 | 0.5187 | 0.1371 | 0.3 | 0.1882 |
| Llama3.2-3B-Instruct + SFT | **0.6793** | 0.3598 | 0.4704 | 0.3424 | 0.3485 | 0.3454 |
| Llama3.2-3B-Instruct + SFT + RS-DPO | 0.654 | **0.6468** | **0.6504** | **0.491** | **0.4788** | **0.4848** |

Table 18: **Cross-Idiom Generalization on Python Ruff Idioms:** We evaluate the effect of different METALINT training setups (SFT, RS-SFT, and RS-DPO) on Qwen3-4B (with and without reasoning) and Llama3.2-3B. Models are trained on easy synthetic Python Ruff idioms and tested on other Ruff idioms with varying levels of transfer (section 4.2). Best score across the compared training setups per model are bolded.

| Model | Detection | | | Localization | | |
|---|---|---|---|---|---|---|
| | $P_{Det}$ | $R_{Det}$ | $F_{Det}$ | $P_{Loc}$ | $R_{Loc}$ | $F_{Loc}$ |
| Qwen3-4B | 0.5267 | 0.1715 | 0.2587 | 0.0954 | 0.0824 | 0.0884 |
| Qwen3-4B + SFT | 0.4333 | 0.0821 | 0.1381 | 0.0432 | 0.0221 | 0.0292 |
| Qwen3-4B + SFT + RS-DPO | **0.7031** | **0.7043** | **0.7037** | **0.3536** | **0.193** | **0.2497** |
| Qwen3-4B w CoT | 0.8154 | 0.3986 | 0.5354 | 0.2625 | 0.1467 | 0.1882 |
| Qwen3-4B w CoT + RS-SFT | 0.7615 | 0.3689 | 0.497 | 0.2785 | 0.1437 | 0.1896 |
| Qwen3-4B w CoT + RS-SFT + RS-DPO | **0.9303** | **0.4958** | **0.6468** | **0.3482** | **0.2169** | **0.2673** |
| Llama3.2-3B-Instruct | **0.7042** | 0.214 | 0.3283 | 0.0691 | 0.0798 | 0.0741 |
| Llama3.2-3B-Instruct + SFT | 0.5627 | 0.259 | 0.3547 | 0.1066 | 0.0509 | 0.0689 |
| Llama3.2-3B-Instruct + SFT + RS-DPO | 0.6368 | **0.5614** | **0.5965** | **0.2364** | **0.1263** | **0.1647** |

Table 19: **Easy-to-Hard Generalization on PEP Idioms:** We evaluate the effect of different METALINT training setups (SFT, RS-SFT, and RS-DPO) on Qwen3-4B (with and without reasoning) and Llama3.2-3B. Models are trained on easy synthetic Python Ruff idioms and tested on hard manually curated PEP idiom detection data which can't be handled by linters or static analyzers (section 4.3). Best score across the compared training setups per model are bolded.

| Model Comparison | Detection | Localization P | Localization R |
|---|---|---|---|
| Qwen3-4B vs o3-mini | 1266.5 (7.20e-21) | 743.5 (2.96e-11) | 739.0 (4.23e-09) |
| Qwen3-4B w CoT vs o3-mini | 921.5 (3.23e-09) | 2385.5 (9.61e-02) | 1891.0 (4.99e-04) |

Table 20: Wilcoxon signed-rank test results comparing untrained Qwen3-4B variants with `o3-mini`, using Bonferroni-adjusted significance threshold $\alpha = 0.025$. Each cell reports the test statistic (p-value).

| Model Comparison | Detection | Localization P | Localization R |
|---|---|---|---|
| METALINT (SFT) vs METALINT (SFT+RS-DPO) | 7192.0 (1.92e-12) | 0.0 (2.49e-20) | 0.0 (2.92e-18) |
| METALINT w CoT (RS-SFT) vs METALINT w CoT (RS-SFT+RS-DPO) | 839.0 (2.18e-03) | 740.0 (3.95e-03) | 523.0 (2.34e-05) |
| METALINT (SFT) vs METALINT w CoT (RS-SFT) | 528.0 (6.91e-14) | 11.0 (1.38e-15) | 113.0 (7.95e-12) |
| METALINT (SFT+RS-DPO) vs METALINT w CoT (RS-SFT+RS-DPO) | 8140.0 (5.55e-01) | 2568.5 (8.42e-01) | 2544.0 (4.44e-01) |

Table 21: Wilcoxon signed-rank test results comparing MetaLint variants. Each cell reports test statistic (p-value). All the METALINT models are trained Qwen3-4B variants. We use the Bonferroni corrected significance threshold $\alpha = 0.0125$.

| Model Comparison | Detection | Localization P | Localization R |
|---|---|---|---|
| Qwen3-4B vs Qwen3-4B METALINT (SFT) | 560.5 (8.64e-03) | 363.5 (1.82e-02) | 238.0 (4.85e-04) |
| Qwen3-4B vs Qwen3-4B METALINT (SFT+RS-DPO) | 7260.0 (4.13e-09) | 411.0 (1.99e-15) | 979.5 (7.30e-09) |
| Qwen3-4B w CoT vs Qwen3-4B METALINT w CoT (RS-SFT) | 1224.0 (7.22e-01) | 937.0 (6.12e-01) | 918.5 (5.39e-01) |
| Qwen3-4B w CoT vs Qwen3-4B METALINT w CoT (RS-SFT+RS-DPO) | 1728.0 (1.83e-02) | 1011.0 (1.53e-03) | 966.0 (1.02e-03) |

Table 22: Wilcoxon signed-rank test results comparing METALINT models against their untrained counterparts, with Bonferroni-adjusted significance threshold $\alpha = 0.0125$. Each cell reports the test statistic (p-value).

| Model Comparison | Detection | Localization P | Localization R |
|---|---|---|---|
| Qwen3-8B vs METALINT (SFT+RS-DPO) | 8140.5 (7.21e-03) | 1309.5 (2.19e-08) | 2067.0 (2.12e-03) |
| Qwen3-8B w CoT vs METALINT w CoT (RS-SFT+RS-DPO) | 2070.0 (9.17e-01) | 1974.5 (1.88e-01) | 2161.0 (6.58e-01) |
| Qwen3-14B vs METALINT (SFT+RS-DPO) | 7304.0 (4.96e-01) | 2816.5 (3.15e-02) | 3159.0 (2.88e-02) |
| Qwen3-14B w CoT vs METALINT w CoT (RS-SFT+RS-DPO) | 2392.0 (2.78e-01) | 2749.5 (1.26e-01) | 2319.0 (1.32e-03) |
| Qwen3-32B vs METALINT (SFT+RS-DPO) | 7175.0 (4.48e-01) | 3262.0 (1.93e-02) | 2818.5 (5.56e-03) |
| Qwen3-32B w CoT vs METALINT w CoT (RS-SFT+RS-DPO) | 2677.5 (9.95e-03) | 3479.0 (8.38e-02) | 3180.5 (5.64e-04) |
| R1-Distill-Qwen-7B vs METALINT (SFT+RS-DPO) | 8244.0 (2.65e-08) | 555.0 (8.12e-15) | 1924.0 (1.91e-04) |
| R1-Distill-Qwen-7B vs METALINT w CoT (RS-SFT+RS-DPO) | 2907.0 (7.07e-10) | 915.5 (1.04e-12) | 1569.5 (8.64e-06) |
| R1-Distill-Qwen-14B vs METALINT (SFT+RS-DPO) | 9877.0 (3.99e-06) | 2582.5 (9.36e-06) | 3085.0 (2.00e-03) |
| R1-Distill-Qwen-14B vs METALINT w CoT (RS-SFT+RS-DPO) | 2660.0 (9.11e-08) | 1703.0 (2.98e-06) | 1791.0 (3.97e-05) |
| R1-Distill-Qwen-32B vs METALINT (SFT+RS-DPO) | 8677.5 (2.51e-01) | 5767.5 (1.93e-01) | 3705.0 (6.67e-06) |
| R1-Distill-Qwen-32B vs METALINT w CoT (RS-SFT+RS-DPO) | 3125.0 (3.11e-02) | 3641.5 (6.35e-02) | 2175.0 (4.99e-06) |
| Qwen2.5-3B vs METALINT (SFT+RS-DPO) | 8001.0 (1.41e-15) | 0.0 (1.24e-22) | 68.5 (1.26e-19) |
| Qwen2.5-3B vs METALINT w CoT (RS-SFT+RS-DPO) | 949.0 (4.96e-23) | 0.0 (4.24e-22) | 0.0 (9.89e-20) |
| Qwen2.5-7B vs METALINT (SFT+RS-DPO) | 7312.5 (3.37e-10) | 208.0 (1.44e-18) | 610.0 (1.99e-12) |
| Qwen2.5-7B vs METALINT w CoT (RS-SFT+RS-DPO) | 1187.5 (1.14e-14) | 226.0 (2.60e-18) | 406.5 (5.69e-14) |
| Qwen2.5-14B vs METALINT (SFT+RS-DPO) | 8677.5 (2.51e-01) | 3045.5 (5.70e-04) | 4006.0 (3.83e-01) |
| Qwen2.5-14B vs METALINT w CoT (RS-SFT+RS-DPO) | 4123.0 (4.86e-01) | 3228.0 (1.51e-03) | 4383.0 (9.89e-01) |
| Qwen2.5-32B vs METALINT (SFT+RS-DPO) | 8640.0 (1.76e-04) | 1492.5 (3.12e-08) | 2971.5 (4.55e-02) |
| Qwen2.5-32B vs METALINT w CoT (RS-SFT+RS-DPO) | 1792.0 (8.16e-06) | 1166.0 (2.43e-08) | 1983.5 (4.71e-03) |
| Qwen2.5Coder-3B vs METALINT (SFT+RS-DPO) | 11184.0 (8.64e-03) | 953.5 (3.73e-12) | 1716.0 (3.00e-07) |
| Qwen2.5Coder-3B vs METALINT w CoT (RS-SFT+RS-DPO) | 3683.5 (6.45e-03) | 1126.0 (4.39e-11) | 1403.5 (1.32e-08) |
| Qwen2.5Coder-7B vs METALINT (SFT+RS-DPO) | 8001.0 (1.41e-15) | 0.0 (7.69e-23) | 0.0 (2.02e-20) |
| Qwen2.5Coder-7B vs METALINT w CoT (RS-SFT+RS-DPO) | 949.0 (4.96e-23) | 0.0 (2.61e-22) | 0.0 (6.55e-20) |
| Qwen2.5Coder-14B vs METALINT (SFT+RS-DPO) | 9123.5 (1.04e-12) | 289.0 (8.19e-20) | 736.0 (7.32e-14) |
| Qwen2.5Coder-14B vs METALINT w CoT (RS-SFT+RS-DPO) | 1112.0 (1.82e-19) | 159.5 (7.63e-20) | 408.5 (3.02e-15) |
| Qwen2.5Coder-32B vs METALINT (SFT+RS-DPO) | 6833.5 (2.86e-01) | 4500.0 (9.59e-01) | 2651.5 (7.07e-05) |
| Qwen2.5Coder-32B vs METALINT w CoT (RS-SFT+RS-DPO) | 1039.5 (1.16e-02) | 2655.0 (9.39e-01) | 1235.5 (1.44e-05) |
| o3-mini vs METALINT (SFT+RS-DPO) | 7520.0 (4.83e-02) | 4986.0 (5.20e-01) | 4427.5 (2.87e-01) |
| o3-mini vs METALINT w CoT (RS-SFT+RS-DPO) | 1944.0 (7.15e-04) | 3169.0 (5.16e-01) | 2683.0 (4.23e-01) |

Table 23: Wilcoxon signed-rank test statistics and p-values comparing MetaLint variants against baseline models. All the METALINT variants are Qwen3-4B variants and Qwen2.5 and Qwen2.5Coder variants are instruction tuned checkpoints. We use the Bonferroni corrected significance threshold $\alpha = 0.0017$.

| Model Comparison | Detection | Localization P | Localization R |
|---|---|---|---|
| Qwen3-4B vs Qwen3-4B w CoT | 1260.0 (3.99e-08) | 425.0 (1.91e-09) | 624.5 (2.25e-05) |
| Qwen3-8B vs Qwen3-8B w CoT | 1924.0 (4.27e-03) | 1132.5 (8.69e-06) | 1005.0 (1.15e-04) |
| Qwen3-14B vs Qwen3-14B w CoT | 1691.0 (2.01e-01) | 2127.0 (4.27e-04) | 2398.5 (1.26e-01) |
| Qwen3-32B vs Qwen3-32B w CoT | 1572.5 (2.75e-01) | 1767.0 (6.35e-06) | 2596.5 (7.31e-02) |

Table 24: Wilcoxon signed-rank test results measuring the effect of Chain-of-Thought (CoT) prompting across Qwen3 model scales. Each cell reports test statistic (p-value). We use the Bonferroni corrected significance threshold $\alpha = 0.0125$.

| Model Comparison | Detection | Localization P | Localization R |
|---|---|---|---|
| Qwen3-4B vs Qwen3-8B | 546.0 (2.35e-08) | 618.0 (1.37e-04) | 572.5 (1.72e-03) |
| Qwen3-8B vs Qwen3-14B | 1350.0 (2.11e-03) | 1503.5 (3.96e-05) | 1008.5 (2.69e-07) |
| Qwen3-14B vs Qwen3-32B | 875.0 (2.22e-02) | 3129.5 (7.94e-01) | 2061.0 (4.14e-01) |
| Qwen3-4B w CoT vs Qwen3-8B w CoT | 1468.5 (1.90e-02) | 1578.5 (1.07e-01) | 1081.5 (2.60e-03) |
| Qwen3-8B w CoT vs Qwen3-14B w CoT | 904.5 (1.40e-01) | 1248.0 (1.66e-03) | 1099.5 (1.82e-05) |
| Qwen3-14B w CoT vs Qwen3-32B w CoT | 850.0 (3.78e-02) | 1834.5 (4.87e-01) | 2396.0 (4.40e-01) |
| R1-Distill-Qwen-7B vs R1-Distill-Qwen-14B | 4278.0 (2.67e-01) | 1431.0 (4.66e-03) | 1962.5 (6.01e-01) |
| R1-Distill-Qwen-14B vs R1-Distill-Qwen-32B | 2432.0 (1.44e-12) | 1475.5 (8.07e-11) | 843.5 (6.38e-15) |
| Qwen2.5Coder-3B vs Qwen2.5Coder-7B | 1541.0 (4.56e-14) | 0.0 (3.46e-10) | 0.0 (7.07e-10) |
| Qwen2.5Coder-7B vs Qwen2.5Coder-14B | 8.0 (7.89e-04) | 0.0 (2.04e-03) | 0.0 (2.14e-03) |
| Qwen2.5Coder-14B vs Qwen2.5Coder-32B | 423.0 (2.83e-27) | 100.5 (2.00e-22) | 43.5 (9.29e-23) |
| Qwen2.5-3B vs Qwen2.5-7B | 18.0 (2.43e-08) | 0.0 (3.43e-05) | 0.0 (3.58e-05) |
| Qwen2.5-7B vs Qwen2.5-14B | 960.5 (1.70e-13) | 500.5 (1.65e-11) | 574.0 (2.62e-10) |
| Qwen2.5-14B vs Qwen2.5-32B | 1925.0 (1.87e-04) | 1858.5 (6.60e-03) | 2108.5 (1.43e-02) |

Table 25: Wilcoxon signed-rank test results measuring the effect of increasing model scale across families and CoT settings. Each cell shows the test statistic (p-value). All Qwen2.5 and Qwen2.5Coder variants are instruction tuned checkpoints. We use the Bonferroni corrected significance threshold $\alpha = 0.0036$.

| Model Comparison | Detection | Localization P | Localization R |
|---|---|---|---|
| GPT-4o vs GPT-4.1 | 2550.0 (9.21e-01) | 3961.0 (5.83e-01) | 2207.5 (4.48e-06) |
| o3-mini vs o4-mini | 575.0 (6.68e-01) | 1171.0 (1.44e-03) | 1079.0 (1.47e-04) |

Table 26: Wilcoxon signed-rank test results comparing GPT model variants. Each cell shows the test statistic (p-value). We use the Bonferroni corrected significance threshold $\alpha = 0.025$.

| PEP | Chain of Thought | Reason for Failure |
|---|---|---|
| 614 | ... I need to check the code file for violations of the 614 idiom, which is about using complex expressions in decorators. ... 
 ... The register variable is defined on line 7 as partial(register, server=hippa). So, the decorator is a partial function. ... 
 ... the idiom allows for complex expressions in the decorator. For example, if the decorator was something like @lambda func: timed(...)(func), that would be allowed. ... 
 ... the register is a partial function, which is a simple expression. ... 
 ... the decorator is just a variable that's a partial function. ... | Ambiguity about what constitutes a complex expression. The idiom targets decorators with complex expressions that make code more verbose |
| 525 | ... The user ... wants me to check for ... idiom, 525, which is about using asynchronous generators with "async for" syntax ... the violation would be if the code uses synchronous generators (like regular for loops) instead of asynchronous generators (using async for) ... Wait, the example given in the idiom is a class that uses __aiter__ and __anext__ to create an asynchronous iterator. The fix is to use a function with async def and yield. So, I need to check if there are any instances where the code is using synchronous generators (like for loops) instead of asynchronous ones. ... Looking through the code, I don't see any for loops that are using 'async for'. ... | CoT accidentally looks for the incorrect construct at a crucial point (async for instead of for) |
| 487 | ... using __init_subclass__ and __set_name__ for class customization without metaclasses ... using metaclasses for simple customization tasks makes code unnecessarily complex ... there's a class called FirestoreClientMeta, which is a metaclass ... used for registration of transports ... the problem is whether this is a simple customization ... not just for a single customization but for managing multiple transports ... the idiom is about using metaclasses for simple customization ... but in this case ... handling a more complex task ... maybe this is not a violation ... | Ambiguity around what constitutes a simple class. The idiom targets instances where metaclasses are used for simple customization tasks that could have been handled with "__init_subclass__" or "__set_name__") |

Table 27: Example chains of thought for various PEPs where the CoT model incorrectly flags NO VIOLATIONS FOUND instead of the non CoT model.

| Model | Transfer | Detection | | | Localization | | |
|---|---|---|---|---|---|---|---|
| | | $P_{Det}$ | $R_{Det}$ | $F_{Det}$ | $P_{Det}$ | $R_{Det}$ | $F_{Det}$ |
| Llama3.2-3B-Instruct | | 0.0457 | 0.0079 | 0.0134 | 0.0015 | 0.0022 | 0.0017 |
| Llama3.2-3B-Instruct + SFT | PMD → PMD | 0.2251 | 0.4421 | 0.2983 | 0.2822 | 0.2778 | 0.28 |
| Llama3.2-3B-Instruct + SFT + RS-DPO | | 0.4395 | 0.8908 | 0.5886 | 0.593 | 0.5969 | 0.5949 |
| Llama3.1-8B-Instruct | | 0.3656 | 0.4015 | 0.3827 | 0.1253 | 0.131 | 0.1281 |
| Llama3.1-8B-Instruct + SFT | PMD → PMD | 0.2264 | 0.4508 | 0.3014 | 0.3201 | 0.3152 | 0.3177 |
| Llama3.1-8B-Instruct + SFT + RS-DPO | | 0.4427 | 0.9191 | 0.5976 | 0.6506 | 0.6709 | 0.6606 |
| Llama3.2-3B-Instruct | | 0.3855 | 0.0096 | 0.0187 | 0.0005 | 0.0004 | 0.0005 |
| Llama3.2-3B-Instruct + SFT | PMD → JEP | 0.2286 | 0.4072 | 0.2928 | 0.1626 | 0.1336 | 0.1467 |
| Llama3.2-3B-Instruct + SFT + RS-DPO | | 0.4903 | 0.8338 | 0.6175 | 0.4216 | 0.3333 | 0.3721 |
| Llama3.1-8B-Instruct | | 0 | 0 | 0 | 0 | 0 | 0 |
| Llama3.1-8B-Instruct + SFT | PMD → JEP | 0.2166 | 0.3724 | 0.2739 | 0.1455 | 0.1142 | 0.128 |
| Llama3.1-8B-Instruct + SFT + RS-DPO | | 0.4964 | 0.8047 | 0.614 | 0.4615 | 0.3395 | 0.3912 |
| Llama3.2-3B-Instruct | | 0.3855 | 0.0096 | 0.0187 | 0.0005 | 0.0004 | 0.0005 |
| Llama3.2-3B-Instruct + SFT | JEP → JEP | 0.9567 | 0.8411 | 0.8952 | 0.7837 | 0.754 | 0.7686 |
| Llama3.2-3B-Instruct + SFT + RS-DPO | | 0.9406 | 0.86 | 0.8985 | 0.7859 | 0.7651 | 0.7753 |
| Llama3.1-8B-Instruct | | 0 | 0 | 0 | 0 | 0 | 0 |
| Llama3.1-8B-Instruct + SFT | JEP → JEP | 0.9658 | 0.8466 | 0.9023 | 0.809 | 0.7844 | 0.7965 |
| Llama3.1-8B-Instruct + SFT + RS-DPO | | 0.9308 | 0.8686 | 0.8986 | 0.8131 | 0.7756 | 0.7939 |
| Llama3.2-3B-Instruct | | 0.0457 | 0.0079 | 0.0134 | 0.0015 | 0.0022 | 0.0017 |
| Llama3.2-3B-Instruct + SFT | JEP → PMD | 0.3722 | 0.2708 | 0.3152 | 0.0574 | 0.0869 | 0.0692 |
| Llama3.2-3B-Instruct + SFT + RS-DPO | | 0.4322 | 0.4054 | 0.4183 | 0.0878 | 0.1222 | 0.1022 |
| Llama3.1-8B-Instruct | | 0.3656 | 0.4015 | 0.3827 | 0.1253 | 0.131 | 0.1281 |
| Llama3.1-8B-Instruct + SFT | JEP → PMD | 0.3514 | 0.2229 | 0.2728 | 0.0383 | 0.0753 | 0.0508 |
| Llama3.1-8B-Instruct + SFT + RS-DPO | | 0.436 | 0.4898 | 0.4613 | 0.0831 | 0.1351 | 0.1029 |

Table 28: **Cross-Idiom Generalization on JEP & PMD Idioms:** Effect of different METALINT training setups (SFT and RS-DPO) on Llama3.2-3B-Instruct (Table 28). The transfer column indicates training and test data on the left and right side of the arrow. Best score across the compared training setups per model are bolded.

