# OpenReview forum: "MetaLint: Generalizable Idiomatic Code Quality Analysis Through Instruction-Following and Easy-to-Hard Generalization"
_ICLR.cc/2026/Conference — Submitted to ICLR 2026_

### Official Review · Reviewer_TK22 · 2025-10-20

**Soundness:** 3
**Presentation:** 2
**Contribution:** 2
**Rating:** 4
**Confidence:** 3

**Summary:**

This paper proposes METALINT, a training framework for generalizable code quality analysis. The core problem it addresses is the issue of outdated rules that arises when using large models for code quality checks, as these models cannot adapt to new rules. Inspired by the generalization capabilities of instruction tuning, the paper constructs data pairs consisting of rule descriptions, code, and the locations of rule violations. This approach guides the model to learn rules from their descriptions rather than relying on memory, thereby enabling it to  generalize to new rules.
The training process employs a two-stage SFT+DPO method, which standardizes the model's output while enhancing its generalization ability. Finally, the framework was tested on models of various scales, yielding mixed results. On the positive side, a small model achieved a violation detection recall that surpassed even top-tier large models. On the negative side, SFT training alone led to a performance decline, and all models performed very poorly in accurately localizing the line numbers of violations.

**Strengths:**

1. The paper convincingly articulates the limitations of current linter/static analysis tools and LLM-based code quality checkers, emphasizing the challenge of evolving/crossing idiom boundaries and the pitfalls of memorization.
2. The experiments confirm the method's effectiveness, though a gap remains when compared to top-tier general-purpose models.
3. The work is expected to have a notable impact in the field of code quality analysis.

**Weaknesses:**

1. The paper is missing the section “The Use of Large Language Models (LLMs)”.

2.While METALINT leverages existing linters (e.g., Ruff, PMD) as verifiers in preference optimization, this design partially contradicts its core motivation—moving beyond static rule-based systems. The reliance on linters for feedback might limit the semantic depth of preference learning, especially for complex idioms where linters are inherently insufficient. A stronger or human-aligned verification mechanism would strengthen the claim of adaptive, semantics-aware generalization.

3.While the paper claims to achieve “easy-to-hard generalization”, the proposed framework does not include any mechanism explicitly designed to enhance generalization. The observed transfer ability largely stems from the intrinsic properties of DPO, which is already known to improve alignment and robustness in instruction-tuned LLMs. In other words, the reported generalization reflects DPO’s inherent strengths rather than any algorithmic innovation introduced by METALINT itself.

4.The figures are too rough, posing obstacles to reading the paper. For example, the flow arrows in the verifiable reward diagram in Figure 1 are confusing. The rows and columns in the result tables are also misaligned, for example, Detection and Localization in Table 1.

5. $\mathcal{L}$ is usually regarded as a loss function and is not recommended for use in linters.

**Questions:**

1.Why does using CoT on the base model not cause a drop in recall, while using CoT on the model after the two-stage SFT-DPO training causes a sharp drop in recall?

2.What is the purpose of SFT? The experimental results show that adding SFT caused a decline in all three metrics. For a model that has already been instruction-tuned, is this kind of SFT redundant and does it damage the model's generalization ability?

---

> ### Author Response · Authors · 2025-11-25
> **Response to Reviewer TK22**
>
> We thank the reviewer for their time and feedback, and respond to the weaknesses (Wi) and questions (Qi) below:
>
> **W1:** We apologize for the oversight and thank the reviewer for pointing this out, and will add this section in the revised version of this paper.
>
> **W2:** We understand the concern that using linters as verifiers may appear at odds with the goal of moving beyond static rule-based systems. However, this design choice is intentional. It allows us to test whether the easy-to-hard generalization effect observed in prior work on mathematical reasoning [1] also holds in a code quality setting, even when the feedback source is limited. The key research question is whether preference optimization can extrapolate from simple, rule-based signals to harder, semantics-aware phenomena that linters cannot capture. Our results show that this effect does transfer to this new domain, which is a nontrivial empirical finding and the primary motivation for our design.
>
> **W3:** We acknowledge the reviewer’s point regarding the lack of novelty in the RS-DPO algorithm itself. However, the key contribution of our work lies in the task framing. We cast code-quality analysis as an instruction-following task, incorporating best-practice guidelines into the input. This allows the guidelines to be modified or replaced at test time with more nuanced or hard-to-apply rules, which enables easy-to-hard generalization when combined with DPO. While DPO’s inherent strengths facilitate alignment, without this instruction-following framing, the model would be limited to flagging only practices seen during training, suffering from the same static limitations as prior work [2,3]. Our contribution is thus in demonstrating that this task framing allows DPO to generalize in a domain where such behavior had not been observed before.
>
> **W4, W5:** We thank the reviewer for pointing out these issues and will fix them in the revised version of the paper. We will also redesign the figures and make them simpler and easier to follow, and edit the text to explain them better.
>
> **Q1:** We believe the reviewer may have misunderstood the experimental setup. Although we toggle CoT on or off for the base Qwen3-4B model, the two models compared after the two-stage training are not the same checkpoint. The CoT and non-CoT variants, though initialized from the same base model, diverge after the SFT stage (CoT setting is trained with RS-SFT while the non-CoT setting is trained with SFT).
>
> For the CoT condition, we start from the base Qwen3-4B-CoT model (CoT is toggled on, but the same set of weights as Qwen3-4B), apply RS-SFT to obtain Qwen3-4B-CoT + RS-SFT, and then apply RS-DPO to obtain Qwen3-4B-CoT + RS-SFT + RS-DPO. For the non-CoT condition, we again begin with the base Qwen3-4B model, apply SFT on linter-derived labels to obtain Qwen3-4B + SFT, and then apply RS-DPO to obtain Qwen3-4B + SFT + RS-DPO.
>
> Both final models outperform their respective baselines (Qwen3-4B and Qwen3-4B-CoT). However, Qwen3-4B + SFT + RS-DPO shows higher recall and lower precision, while Qwen3-4B-CoT + RS-SFT + RS-DPO shows the opposite pattern. These differences arise because they are distinct checkpoints with different training signals, not because adding CoT after two stage training degrades recall.
>
> **Q2:** As noted in Section 5.2 (lines 402 to 406), training Qwen3-4B (CoT) with RS-DPO without SFT led to near-zero performance because many generated preference pairs violated the required output format, leading to this bad behavior being learned during RS-DPO. Although SFT slightly reduces downstream metrics, it is necessary to teach format compliance, create high-quality data for RS-DPO, and enable the subsequent easy-to-hard generalization effect. Thus, SFT is not redundant even for an instruction-tuned base model, since it establishes the consistent output structure that RS-DPO relies on.
>
> [1] Sun, Zhiqing, et al. "Easy-to-hard generalization: Scalable alignment beyond human supervision." Advances in Neural Information Processing Systems 37 (2024): 51118-51168.
> [2] Holden, Darren, and Nafiseh Kahani. "Code linting using language models." arXiv preprint arXiv:2406.19508 (2024).
> [3] Vijayvergiya, Manushree, et al. "Ai-assisted assessment of coding practices in modern code review." Proceedings of the 1st ACM International Conference on AI-Powered Software. 2024.

---

### Official Review · Reviewer_cMmU · 2025-10-27

**Soundness:** 2
**Presentation:** 3
**Contribution:** 2
**Rating:** 4
**Confidence:** 4

**Summary:**

This paper presents an approach to fine-tuning LLMs to detect and localize programming idioms in programs. The basic idea is to construct a dataset using rule-based techniques to detect programming idioms in existing source code. Based on the dataset, SFT and RL are used to fine-tune LLMs to acquire the capability of detecting and localizing programming idioms in programs. The fine-tuned LLM (especially the LLM fine-tuned with RL) is superior to the base LLM, and even for harder benchmark, the superiority of the fine-tuned LLM over the base LLM exists.

**Strengths:**

1. The idea of obtaining a strong code quality analyzer via fine-tuning on a dataset of easier instances is interesting.
2. The empirical results indicate that SFT and RL are effective for training a code quality analyzer.

**Weaknesses:**

1. The current evidence may not support that the obtained LLM-based code quality analyzer is a stronger one. Since the base LLM is not specific to code quality analysis, the empirical superiority over the base LLM does not suffice to claim a stronger code quality analyzer. At least, the fine-tuned LLM-based code quality analyzer should be compared with the rule-based approaches that generate the data for fine-tuning.
2. There is lack of empirical comparison with existing SOTA LLM-based (or even deep-learning-based) code quality analyzer. Without such comparison, it is unclear whether the proposed scheme is a better one to obtain a strong code quality analyzer.
3. The novelty seems to be limited. Although the idea of training a analyzer to tackle harder idioms with simple idioms is interesting, this paper seems to rely solely on existing techniques to achieve this purpose.

**Questions:**

1. Can the trained code quality analyzer outperform the rule-based analyzers that generating the the training dataset?
2. How does the trained code quality analyzer compare with SOTA LLM-based code quality analyzers?

---

> ### Author Response · Authors · 2025-11-25
> **Response to Reviewer cMmU**
>
> We thank the reviewer for their time and feedback, and respond to the weaknesses (Wi) and questions (Qi) below:
>
> **W1:** We explicitly compare the trained LLM against the rule-based tools that generated the fine-tuning data. All ground truth labels in our synthetic datasets come directly from Ruff, PMD, and JEP Tree Sitter. In other words, the LLM is evaluated on the same outputs produced by these tools, so any performance gap reflects its inability to perfectly reproduce their results (see PMD→PMD and JEP→JEP transfer rows in Table 28 and in-domain Ruff performance in Table 16). While the LLM comes close in some cases, scores remain far from perfect, particularly for PMD, on detection precision and all localization metrics. As noted in the Conclusion (line 482), "For mechanically easy idioms, linters remain cost-effective, but MetaLint enables detection of abstract idioms, supporting personalized, evolving code quality standards." This shows that our approach complements existing linters, capturing nuanced or hard-to-detect practices that rule-based systems cannot scale to. Additionally, MetaLint provides personalization by adapting to new best practices, which prior works [1, 2, 3, 4] cannot. We welcome suggestions for additional baselines or model checkpoints for comparison, but the key point remains that our method provides capabilities beyond these existing techniques.
>
> **W2:** We tried to compare our approach against prior LLM-based code-quality methods, such as [1] and [4]. While [1] is from Google, the models are not publicly available. [4] trains a CodeBERT model for method-level detection, but due to its 512-token limit (page 10), it cannot handle multi-line files or multiple methods in our setting. Other works, such as [2], propose rule-based linters, and [3] uses a hybrid prompting approach without any training, relying on GPT-3.5-Turbo.
>
> **W3:** We acknowledge that our work does not introduce a new machine learning algorithm. The novelty lies in the task framing: we cast code-quality analysis as an instruction-following task, where best-practice guidelines are part of the input. This design allows guidelines to be modified or replaced at test time with more nuanced or hard-to-apply rules, enabling easy-to-hard generalization when combined with DPO. Without this framing, the model would be limited to flagging only practices seen during training, suffering from the static limitations of prior work [1,4]. Our contribution is demonstrating that this instruction-following framing allows DPO to generalize in code quality analysis, a domain where such behavior has not been observed before our work.
>
> **Q1:** Please refer to our response to W1
>
> **Q2:** Please refer to our response to W2
>
> [1] Vijayvergiya, Manushree, et al. "Ai-assisted assessment of coding practices in modern code review." Proceedings of the 1st ACM International Conference on AI-Powered Software. 2024.
> [2] Zhang, Zejun, et al. "Refactoring to pythonic idioms: A hybrid knowledge-driven approach leveraging large language models." Proceedings of the ACM on Software Engineering 1.FSE (2024): 1107-1128.
> [3] Zhang, Zejun, et al. "Automated refactoring of non-idiomatic python code with pythonic idioms." IEEE Transactions on Software Engineering (2024).
> [4] Holden, Darren, and Nafiseh Kahani. "Code linting using language models." arXiv preprint arXiv:2406.19508 (2024).

---

> > ### Comment · Reviewer_cMmU · 2025-11-28
> > **Respnose to the authors' rebuttal**
> >
> > W1: I saw the sentence in Conclusion (line 482): "For mechanically easy idioms, linters remain cost-effective, but MetaLint enables detection of abstract idioms, supporting personalized, evolving code quality standards." But I cannot find quantitative results supporting this claim in the paper.
> >
> > W2: I think it's the authors' responsibility to provide evidence that the paper does propose a stronger code quality analyzer. The difficulty in performing empirical comparison with SOTA approaches may not be an acceptable reason to avoid this responsibility.

---

> > > ### Author Response · Authors · 2025-11-28
> > > **Follow up to Reviewer cMmU's response**
> > >
> > > We thank the reviewer for engaging with our rebuttal and address their concerns below.
> > >
> > > W1:
> > > 1. The "abstract idioms" referenced in line 482 correspond to the hard PEP idioms (section 4.3, line 303). Quantitative results supporting the claim that "MetaLint enables detection of abstract idioms" are provided in Table 4 and expanded in Table 17, which compares MetaLint-trained models against strong few-shot baselines using state-of-the-art code LLMs.
> > > 2. For claims regarding evolving or personalized coding standards, Section 4.2 presents evidence that MetaLint-trained models generalize from synthetic training idioms (a subset detectable by Ruff) to unseen idioms also detectable by Ruff. We additionally demonstrate cross-linter transfer between JEP and PMD idioms. Models trained on one linter’s idioms improve performance on the other’s unseen idioms relative to the base model. These results collectively show that MetaLint improves a model’s ability to detect idioms not present during training, which is the key requirement for handling evolving or personalized standards.
> > > We will make the following points clearer in the revised version of the manuscript.
> > >
> > > W2:  We acknowledge the reviewer's concern about the strength of MetaLint-trained code quality analyzers. We would like to substantiate the claim that MetaLint leads to a strong code quality analyzer and is stronger than all the previously cited baselines through the following points:
> > >
> > > 1. Linter-based approaches ([2]): For the "hard-PEP" idiom benchmark (Sec. 4.3), we specifically choose code quality standards that cannot be captured by a linter, since they require semantic reasoning about code and developer intent. Hence, by design, MetaLint performs better than any linter based approach like [2].
> > > 2. Embedding based approaches ([4]): The method in [4] targets only single function linting (limited to 512 tokens). MetaLint addresses idiom detection at a file level (with 5510 tokens on average per code file analyzed) and also hard idioms, as mentioned above. In this sense, our formulation extends the scope of [4] and evaluates a superset of the scenarios they consider.
> > > 3. LLM based approaches ([1]): While this paper does train an LLM to do best practice flagging, they do not release code, model checkpoints or data which makes it impossible to include as a baseline.
> > > 4. Hybrid approaches ([3]): This paper combines a linter along with prompting a single closed source model (GPT-3.5-Turbo). As mentioned in point 1, linters by design are unable to tackle the hard PEP benchmark. For LLMs we compare few-shot prompting of 22 strong code LLMs, including 5 frontier OpenAI models such as GPT-5 (high), o3-mini, o4-mini, GPT-4.1 and GPT-4o, all of which are stronger at code generation and reasoning that GPT-3.5-Turbo and our approach outperforms o3-mini on several metrics.
> > >
> > > Hence, we highlight that our baseline comparisons **subsume** prior work and show that MetaLint-trained models perform competitively with fewer (4B) parameters.
> > >
> > > We hope these clarifications address the reviewer's concerns.
> > >
> > > [1] Vijayvergiya, Manushree, et al. "Ai-assisted assessment of coding practices in modern code review." Proceedings of the 1st ACM International Conference on AI-Powered Software. 2024.
> > > [2] Zhang, Zejun, et al. "Refactoring to pythonic idioms: A hybrid knowledge-driven approach leveraging large language models." Proceedings of the ACM on Software Engineering 1.FSE (2024): 1107-1128.
> > > [3] Zhang, Zejun, et al. "Automated refactoring of non-idiomatic python code with pythonic idioms." IEEE Transactions on Software Engineering (2024).
> > > [4] Holden, Darren, and Nafiseh Kahani. "Code linting using language models." arXiv preprint arXiv:2406.19508 (2024).

---

### Official Review · Reviewer_vtVc · 2025-11-01

**Soundness:** 3
**Presentation:** 3
**Contribution:** 2
**Rating:** 6
**Confidence:** 3

**Summary:**

The paper’s core idea is to move “code quality” away from memorizing fixed lint rules and toward instruction-guided detection of code idioms: give the model a natural-language spec plus a few examples, and it flags good/bad idioms in source code and pinpoints the lines. To teach small models to generalize “from easy to hard”, the authors first use real linters (Python Ruff, Java PMD, with a dash of Tree-Sitter) to mass-produce easy idioms for SFT, then apply verifiable preference optimization to boost recall and localization; the reward is simply overlap with the linter’s labeled line set. In experiments on their Hard PEP Idioms benchmark, a Qwen3-4B METALINT model delivers strong detection/localization, especially recall, and even generalizes across model families and over to Java. In short, the contributions are: (1) METALINT, a framework that treats code quality as instruction-following over updatable idioms; (2) a tougher, more realistic evaluation set that goes beyond typical linter coverage; (3) a systematic study showing plain SFT tends to “memorize rules”, while adding DPO meaningfully improves recall and localization; and (4) robust generalization across model sizes, optional CoT, multiple languages, and multiple tools.

**Strengths:**

The paper’s most original contribution is to reframe code quality from fixed rule matching into instruction-conditioned “code idiom” detection, coupled with preference optimization using a verifiable line-level F1 reward. This design reduces the introduction of “new rules or best practices” to supplying a natural-language description plus a few examples, thereby breaking free from the rule engineering and coverage limitations of traditional linters. The authors also build cross-language (Python/Java), cross-model-family, and cross-scale training and evaluation setups to systematically validate the framework’s transfer and extensibility on both detection and localization; in particular, under easy to hard transfer to PEP-level fine-grained idioms, small models still achieve high recall and competitive localization, demonstrating practical potential and scientific value.

The paper’s quality and clarity are also strong: the sources and synthesis process of training data are reproducible; the evaluation metrics distinguish detection from localization and specify formatting constraints and the “no violations” case; and the ablations span SFT vs. DPO, with/without chain-of-thought, and different data compositions. The conclusions consistently indicate that verifiable rewards + preference optimization encourage generalization rather than memorization.

**Weaknesses:**

1. The benchmark feels small and narrowly built. The PEP “hard” set isn’t that big, and the initial collection relies on high-recall heuristics followed by manual cleanup, easy to introduce retrieval bias and hurt external validity. Also, there’s no mention of inter-annotator agreement.

2. New application not equals to new method. Porting instruction tuning, synthetic data, and RS-DPO to this setting is neat, but these are mature techniques. It reads more like a solid deployment to a new domain than a fundamental methodological breakthrough.

3. Missing key baselines. There’s no few-shot training directly on the hard PEP data, no RAG comparison, and no multi-task setup that mixes easy/hard idioms—all of which are natural baselines to include.

4. Localization is weak and under-analyzed. Line-level localization F1 is only 26.73%, and there’s no real error analysis explaining why localization lags detection. It also doesn’t discuss whether coarser localization (e.g., function-level instead of line-level) might be sufficient or more reliable.

**Questions:**

Could you clarify the distinction between “hard idioms” and “unseen idioms”? Is “hard” about intrinsic complexity semantics or context or multi-span localization, or does it simply mean the idiom wasn’t present in training and is therefore “unseen”? Have you tried idioms that are fundamentally different from both Ruff and PEP (e.g., cross-file semantics, resource lifecycles, concurrency patterns) to test robust generalization?

Regarding the ablation on the proportion of “NO VIOLATIONS FOUND” cases((k%)): you report that 5% yields the best F1. Does this hold across different models and datasets, or does it require per-model tuning?

---

> ### Author Response · Authors · 2025-11-25
> **Response to Reviewer vtVc (1/2)**
>
> We thank the reviewer for recognizing the quality, clarity, and reproducibility of our work. We also recognize their concerns about novelty and the size and quality of the hard PEP benchmark and address them below with our responses to the weaknesses (Wi) and questions (Qi).
>
> **W1:** We appreciate the reviewer’s concern regarding the dataset size. However, we respectfully disagree that the benchmark is small. In practice, it is sufficiently large to yield statistically significant differences across the models we compare, and it exposes clear and statistically significant gains between the base model and the MetaLint-trained model. These gains demonstrate effective, easy-to-hard generalization (see Appendix E2).
>
> Regarding retrieval bias in the initial collection step, we agree that this is a potential risk. This is precisely why we design our heuristics to maximize recall. For example, PEP 506 pertains specifically to the random module, so files that do not import or use random are unlikely to contain PEP 506 violations. Our heuristics follow this type of commonsense filtering and are explicitly documented in Tables 13, 14, and 15. We welcome feedback from the reviewer on ways to further improve these heuristics.
>
> We also thank the reviewer for pointing out the missing discussion of inter-annotator agreement. The average Cohen’s κ across annotators is 0.95. To compute this, we converted each annotated range (for example, lines 13 to 15) into per-line binary labels, where a line containing a violation is marked as 1 and all others as 0. Cohen’s κ was then computed over all lines for each file and averaged across the dataset. The high agreement reflects the clarity of the annotation scheme. We will include this analysis in the Appendix of the revised submission.
>
> **W2:** We appreciate the reviewer’s point and will revise the framing of our contributions accordingly. Our work does not propose new learning algorithms; rather, its novelty lies in (1) a scalable synthetic-data pipeline tailored to code-quality idioms, (2) a training approach that combines instruction fine-tuning [1] and preference optimization [2] to enable easy-to-hard generalization in this domain [3], and (3) a benchmark of challenging PEP-derived code-quality concepts that are not detectable by existing linters but expose genuine weaknesses in state-of-the-art LLMs.
>
> Although the underlying techniques are established, applying them to this domain fills concrete gaps in current tool-based, LLM-based, and hybrid approaches, which either cannot incorporate evolving best practices [4], restrict themselves to narrow concerns such as naming or formatting [5, 6], or are trained on a fixed set of quality issues [7]. The instruction-following setup, combined with DPO using a verifiable reward to construct preference pairs, is key to achieving easy-to-hard transfer for these fine-grained code-quality idioms, something prior LLM-based code-quality systems have not demonstrated. Finally, the ablations on the proportion of NO VIOLATIONS instances reveal a clear precision-recall tradeoff within DPO. This behavior is scientifically interesting and suggests broader implications for data selection during preference optimization for other domains.
>
> [1] Wang, Yizhong, et al. "Super-naturalinstructions: Generalization via declarative instructions on 1600+ nlp tasks." arXiv preprint arXiv:2204.07705 (2022).
> [2] Khaki, Saeed, et al. "Rs-dpo: A hybrid rejection sampling and direct preference optimization method for alignment of large language models." arXiv preprint arXiv:2402.10038 (2024).
> [3] Sun, Zhiqing, et al. "Easy-to-hard generalization: Scalable alignment beyond human supervision." Advances in Neural Information Processing Systems 37 (2024): 51118-51168.
> [4] Vijayvergiya, Manushree, et al. "Ai-assisted assessment of coding practices in modern code review." Proceedings of the 1st ACM International Conference on AI-Powered Software. 2024.
> [5] Zhang, Zejun, et al. "Refactoring to pythonic idioms: A hybrid knowledge-driven approach leveraging large language models." Proceedings of the ACM on Software Engineering 1.FSE (2024): 1107-1128.
> [6] Zhang, Zejun, et al. "Automated refactoring of non-idiomatic python code with pythonic idioms." IEEE Transactions on Software Engineering (2024).
> [7] Holden, Darren, and Nafiseh Kahani. "Code linting using language models." arXiv preprint arXiv:2406.19508 (2024).

---

> ### Author Response · Authors · 2025-11-25
> **Response to Reviewer vtVc (2/2)**
>
> **W3:** We do not train on the hard PEP data because our goal is to evaluate how well models adapt to evolving or newly emerging best-practice distributions purely at test time, without any additional training. For the same reason, we do not use a multi-task setup mixing easy and hard idioms. The hard PEP benchmark already supplies a clear description of each best practice along with illustrative examples, so all models, including base Qwen and Llama variants, are effectively evaluated in a few-shot prompting regime. RAG is also unnecessary in this setting. The prompt contains a compact, self-contained summary of the target best practice, and we assume no labelled data beyond this summary and its examples. Under these assumptions, external retrieval would not provide an additional useful signal.
>
> **W4:** We conducted an error analysis on egregious localization failures made by the Qwen3-4B non-CoT model trained with SFT and RS-DPO. We define an egregious localization error as a case where the model predicts a non-empty set of line numbers that is completely non-overlapping with a non-empty ground-truth set. This isolates pure localization errors, since the model detects the presence of a violation but fails to localize it.
>
> Across the full benchmark, we found 78 such cases (14.5%). We manually annotated 50 of them and observed the following recurring error types:
>
> 1. **Adjacent (6/50):** Predicts a line adjacent to the true line, but not the correct one.
>
> 2. **Before/After (2/50):** Misinterprets the idiom and flags a line that follows the best practice rather than the violating line.
>
> 3. **Definition vs Initialization (2/50):** Flags an initialization instead of the definition that actually constitutes the violation.
>
> 4. **Plausible-Looking (23/50):** Flags a superficially plausible but ultimately incorrect line due to surface-level similarity or insufficient nuance.
>
> 5. **Random (14/50):** Picks a seemingly arbitrary line; detection is correct but localization collapses.
>
> 6. **Repetition (3/50):** Repeats the same incorrect line multiple times, suggesting a decoding or sampling artifact.
>
> For the camera-ready version, we plan to expand this analysis, annotate more cases, and add illustrative plots and a table with representative examples for each error category.
>
> **Q1:** As noted in the introduction near lines 71 to 73, “hard idioms” refer to patterns whose structure makes them difficult or infeasible to capture with rule-based linters. These include nuanced cases such as PEP 506, where constructing precise rule-based checks is not practical. In contrast, “unseen idioms” are simply idioms that do not appear in the model’s training distribution. These two axes are independent. In principle, one could have seen hard idioms or unseen hard idioms, depending on how the training data is curated. In our work, all hard idioms are unseen because they were derived from selected PEPs that appear only in the hard PEP evaluation benchmark.
>
> We did not evaluate idioms involving cross-file semantics, resource lifecycles, or concurrency patterns, since the tasks in this paper operate at the file level. However, we are actively exploring repository-level extensions of this benchmark and plan to study generalization in those settings in future work.
>
> **Q2:** The optimal threshold does vary with model family, dataset, etc., as shown in Table 11 and Table 12, for Qwen3-4B with CoT and Llama3.2-3B-Instruct models. However, we were able to reduce the number of ablations based on the insights from the Qwen3-4B non-CoT setting and empirically found that searching for values ranging from 0% to 10% should be likely sufficient for most cases.

---

### Official Review · Reviewer_ZLtP · 2025-11-01

**Soundness:** 3
**Presentation:** 3
**Contribution:** 3
**Rating:** 6
**Confidence:** 4

**Summary:**

This paper presents METALINT, a framework for teaching LLMs to perform code quality analysis by following natural language instructions. Its core "easy-to-hard" generalization idea involves training (via SFT and DPO) on synthetic data from simple linter rules. The authors show this enables a small 4B model to generalize to complex semantic PEP idioms, achieving SOTA detection recall on a custom benchmark.

**Strengths:**

1.  The paper is well-written and addresses the critical problem of LLM-based code quality analysis, rightly identifying that models struggle to adapt to evolving best practices.
2.  The proposed "easy-to-hard" generalization framework is novel and clever. It leverages instruction tuning on linter-generated synthetic data, avoiding the need for expensive manual annotation.
3.  Using the linter itself as a "verifiable reward model" for preference optimization (RS-DPO) is a strong methodological contribution that provides a data-efficient path to generalization.

**Weaknesses:**

1.  The "meta-linting" formulation appears to evaluate only one idiom specification at a time. This is a potential departure from real-world linters, which must check hundreds of rules simultaneously. It would be beneficial to explore how METALINT's performance scales when many idiom specifications are provided in a single prompt.
2.  The paper convincingly shows that DPO enables generalization from "easy" to "hard" idioms, but the underlying why remains an interesting open question. It would be valuable to further investigate what the model is learning: a generalized concept of "code quality," or a more general "instruction-following" capability. Further analysis here could strengthen this compelling hypothesis.
3.  The SOTA claims on the "hard" PEP benchmark are very promising. However, the benchmark's current size (536 examples) is relatively modest. Expanding this benchmark in future work could help further solidify the robustness of these strong results.
4.  The finding that the CoT model underperforms (lower recall) is counter-intuitive. The paper attributes this to "overthinking." An alternative hypothesis worth exploring is that the RS-SFT data collection (filtering for perfect rewards) may have inadvertently trained the model to be overly conservative when facing ambiguity.

**Questions:**

1.  The non-CoT model achieved the highest recall, while the CoT model had much higher precision. Does this suggest a fundamental precision/recall trade-off? Was an ensemble of the two models considered to potentially achieve the best of both?
2.  Table 10 shows a trade-off based on the fraction of "NO VIOLATIONS" (NV) data in DPO training. The 0% NV model had the highest recall on the Ruff test set. What was the performance (P/R/F1) of this 0% NV model on the "hard" PEP benchmark? Is it possible the main SOTA recall claim is actually an underestimate?
3.  The meta-task definition includes both a description ($D_I$) and examples ($E_I$). How sensitive is the model's performance to the quality and quantity of these examples? An ablation on few-shot vs. zero-shot (description only) instructions would be insightful.

---

> ### Author Response · Authors · 2025-11-25
> **Response to Reviewer ZLtP (1/2)**
>
> We thank the reviewer for recognizing the novelty of our contributions and for the thoughtful criticism. We address weaknesses (Wi) and questions (Qi) below:
>
> **W1:** The choice to evaluate one idiom at a time is intentional. In practice, analyses can be run in parallel for each idiom, and since code-quality checks are typically performed offline, there are no latency constraints that require combining many specifications into a single prompt. Additionally, doing one idiom at a time makes the evaluation results cleaner and easier to interpret and avoids any interference effects between similarly defined idioms.
>
> **W2:** We appreciate the reviewer’s recognition that the paper clearly demonstrates generalization from “easy’’ to “hard’’ idioms. We agree that understanding why this happens remains an open question. Our current analysis probes this partly through the Ruff near-transfer setting. As noted in section 4.2 around line 290, these examples resemble the training idioms in surface form but differ in key details. A model relying only on broad patterns would tend to flag them according to the training idioms. Table 16 shows that near-transfer performance does not improve, and sometimes decreases, with SFT, but is consistently and often substantially improved by RS-DPO. We agree that further investigation would strengthen this line of inquiry. A concrete next step is building paired challenge sets where a file contains both the training-idiom violation and the near-transfer violation, and only the instruction prompt changes. This would help clarify whether the model is developing a broader notion of code-quality reasoning or a more general instruction-following capability. These paired cases could also support mechanistic studies, such as probing activations or steering vectors [1]. We will expand on this direction in the future work section.
>
> **W3:** We thank the reviewer for the positive feedback. We agree that expanding the benchmark will further strengthen the evaluation. As part of ongoing work, we are actively growing the benchmark in size, extending it to additional programming languages, and incorporating repository-level context to improve coverage and robustness.
>
> **W4:** We thank the reviewer for the insightful suggestion. We agree that the RS-SFT filtering process might be encouraging conservative behavior under ambiguity, and we plan to investigate this explicitly as an ablation. We also believe another factor contributing to the CoT performance gap is that when the model cannot produce a correct intermediate reasoning trace, we cannot fine-tune it on the linter-provided ground truth response. Methods such as STaR [2] address this by allowing the model to rationalize reasoning traces from an externally given ground truth response. As noted in Appendix A (“Limitations”), our attempts to apply STaR led the model to leak that it had been given the ground-truth answer rather than producing a plausible chain of thought leading to it. We see resolving this issue as an important future direction and plan to explore techniques to make such supervision viable.
>
> **Q1:** We believe the observed precision/recall differences arise from the training setup rather than a fundamental trade-off. The SFT model receives ground-truth feedback directly from the linter, while the RS-SFT model is optimized using feedback on its own generated responses, which can lead to different error profiles. We did explore simple ensembling, but with only two models, there is no natural majority-vote mechanism. We therefore evaluated two alternatives: taking the intersection or the union of the predicted line numbers. Both approaches underperformed the individual models (as shown in the table below), with the intersection strategy performing particularly poorly. If the reviewer has ideas for more effective ensemble strategies, we would be glad to experiment with them.
>
> | Ensembling Strategy                                          | $P_{Det}$      | $R_{Det}$      | $F_{Det}$      | $P_{Loc}$    | $R_{Loc}$    | $F_{Loc}$    |
> |--------------------------------------------------|---------|---------|---------|-------|-------|-------|
> | Qwen3-4B DPO Intersection (non CoT & CoT)        | 0.0667  | 0.0033  | 0.0063  | 0     | 0     | 0     |
> | Qwen3-4B DPO Union (non CoT & CoT)               | 0.6188  | 0.4187  | 0.4995  | 0.1952| 0.1251| 0.1524|
>
> [1] Zou, Andy, et al. "Representation engineering: A top-down approach to ai transparency." arXiv preprint arXiv:2310.01405 (2023).
> [2] Zelikman, Eric, et al. "Star: Bootstrapping reasoning with reasoning." Advances in Neural Information Processing Systems 35 (2022): 15476-15488.

---

> ### Author Response · Authors · 2025-11-25
> **Response to Reviewer ZLtP (2/2)**
>
> Q2: As suggested by the reviewer, we evaluated the Qwen3-4B (no-reasoning) checkpoint trained with 0% NO VIOLATIONS (NV) data. Its full P/R/F detection and localization results on the "hard" PEP benchmark are shown in the table below. This variant achieves even higher recall than the model reported in the main paper, with a moderate reduction in precision. Notably, the precision drop is smaller than the recall gain, resulting in a stronger overall F score. This confirms that our originally reported recall was indeed an underestimate and that our SOTA recall claim is, if anything, further strengthened by this analysis.
>
> | Model                                             | $P_{Det}$ | $R_{Det}$ | $F_{Det}$ | $P_{Loc}$ | $R_{Loc}$ | $F_{Loc}$ |
> |---------------------------------------------------|---------|---------|---------|---------|---------|---------|
> | Qwen3-4B (SFT+RS-DPO)                             | 0.7031  | 0.7043  | 0.7037  | 0.3536  | 0.1930  | 0.2497  |
> | Qwen3-4B (SFT+RS-DPO) (NO VIOLATIONS)             | 0.6658  | 0.8037  | 0.7283  | 0.3797  | 0.2063  | 0.2674  |
> | Qwen3-4B w CoT (RS-SFT + RS-DPO)                  | 0.9303  | 0.4958  | 0.6468  | 0.3482  | 0.2169  | 0.2673  |
>
> Q3: The prompts for the hard PEP benchmark include one to three examples illustrating the before and after code. We ran a simple ablation with and without these examples for the base models, as well as the final MetaLint Qwen3-4B models trained with SFT and DPO. Across all but one setting, the results showed minimal variation. The only notable change was a modest drop in detection performance for the MetaLint CoT-trained model. This drop is attributable to a higher rate of predicting NO VIOLATIONS, indicating that the CoT variant is more conservative and benefits more from the additional examples. Overall, the ablation suggests that the descriptions alone often provide sufficient information for the task.
>
> | Model                                           | Examples? | $P_{Det}$ | $R_{Det}$ | $F_{Det}$ | $P_{Loc}$ | $R_{Loc}$ | $F_{Loc}$ |
> |-------------------------------------------------|-----------|-------------|-------------|-------------|---------|---------|---------|
> | Qwen3-4B                                        | Yes       | 0.5267      | 0.1715      | 0.2587      | 0.0954  | 0.0824  | 0.0884  |
> | Qwen3-4B with CoT                               | Yes       | 0.8154      | 0.3986      | 0.5354      | 0.2625  | 0.1467  | 0.1882  |
> | Qwen3-4B (SFT+RS-DPO)                            | Yes       | 0.7031      | 0.7043      | 0.7037      | 0.3536  | 0.1930  | 0.2497  |
> | Qwen3-4B w CoT (RS-SFT + RS-DPO)                 | Yes       | 0.9303      | 0.4958      | 0.6468      | 0.3482  | 0.2169  | 0.2673  |
> | Qwen3-4B                                        | No        | 0.5556      | 0.1736      | 0.2645      | 0.0802  | 0.0733  | 0.0766  |
> | Qwen3-4B with CoT                               | No        | 0.8304      | 0.3669      | 0.5089      | 0.2572  | 0.1542  | 0.1928  |
> | Qwen3-4B (SFT+RS-DPO)                            | No        | 0.7167      | 0.7268      | 0.7217      | 0.3396  | 0.1909  | 0.2444  |
> | Qwen3-4B w CoT (RS-SFT + RS-DPO)                 | No        | 0.8215      | 0.4249      | 0.5601      | 0.2860  | 0.1983  | 0.2342  |

---

> > ### Comment · Reviewer_ZLtP · 2025-11-28
> >
> > I thank the authors for their response and for conducting the additional experiments, which have successfully addressed most of my concerns. I appreciate the effort put into the rebuttal. Thus, I will maintain my current score.

---

### Author Response · Authors · 2025-11-25
**General Response on Novelty, Baseline Choices, and Methodological Scope**

We thank all reviewers for their time and thoughtful feedback. We especially appreciate reviewers ZLtP and vtVc for highlighting the quality, clarity, and reproducibility of our work, and reviewer TK22 for noting its potential impact on code quality analysis. Several concerns centered on novelty and baseline selection, and we address these here.

The central limitation our work tackles is the rigidity introduced by static training strategies used in prior code-quality LLMs [4, 6]. Our contribution is an approach that blends two established insights: instruction following enables cross-task generalization [1], and preference tuning/RL can support easy-to-hard generalization in math reasoning [3]. We apply these ideas to code quality to achieve two goals: (1) generalization from linter-generated “easy” data to “hard” cases where linters cannot operate (for which we propose a challenging PEP benchmark), and (2) robustness to distributional shifts arising from evolving best practices (evaluated on both synthetic linter-produced data and human-curated PEP benchmark).

We also provide a scientific analysis of the SFT and RS-DPO training recipe [2], showing how each stage contributes to performance, and making task-specific adjustments such as controlling the proportion of NO VIOLATION instances during RS-DPO. We study these effects across model families, programming languages, model sizes, linters, and reasoning/non-reasoning settings. Our results show that the method not only yields models more robust to changing best practices, but also effectively leverages linter-handled cases to improve performance on cases beyond the reach of linters.

Thus, the aim is **not to create the strongest possible code-quality analyzer**, as inferred by reviewer cMmU, but to **demonstrate a general principle: develadaptable LLMs to extend what can be linted**. Because our goal is to develop an approach that does not rely on further re-training to adapt to new data, we do not include training-based baselines suggested by reviewer vtVc or RAG methods. The prompts we provide already contain self-contained best-practice descriptions, so additional external retrieval is not required, while re-training would contradict our setting.

Finally, our baseline selection is intentionally stronger than past work [4, 5, 6], using more capable LLMs in a few-shot setting. This strengthens our claim that a 4B model trained with our method can remain competitive with substantially larger models.

[1] Wang, Yizhong, et al. "Super-naturalinstructions: Generalization via declarative instructions on 1600+ nlp tasks." arXiv preprint arXiv:2204.07705 (2022).
[2] Khaki, Saeed, et al. "Rs-dpo: A hybrid rejection sampling and direct preference optimization method for alignment of large language models." arXiv preprint arXiv:2402.10038 (2024).
[3] Sun, Zhiqing, et al. "Easy-to-hard generalization: Scalable alignment beyond human supervision." Advances in Neural Information Processing Systems 37 (2024): 51118-51168.
[4] Vijayvergiya, Manushree, et al. "Ai-assisted assessment of coding practices in modern code review." Proceedings of the 1st ACM International Conference on AI-Powered Software. 2024.
[5] Zhang, Zejun, et al. "Automated refactoring of non-idiomatic python code with pythonic idioms." IEEE Transactions on Software Engineering (2024).
[6] Holden, Darren, and Nafiseh Kahani. "Code linting using language models." arXiv preprint arXiv:2406.19508 (2024).

---

### Meta-Review · Area_Chair_JPzZ · 2026-01-03

**Summary:**

The reviews consistently raised concerns about (i) methodological novelty, (ii) whether the approach truly moves beyond static rule-based systems given its reliance on linters as verifiers, (iii) whether the claimed easy-to-hard generalization is convincingly established as a mechanism rather than a repackaging of known effects of instruction-tuning/DPO, and (iv) baseline and evaluation choices that make it difficult to conclude the approach yields a meaningfully stronger or more semantically grounded code-quality analyzer. Overall, despite solid engineering and some promising results, I do not find the contribution sufficiently compelling for acceptance at ICLR.

**Reviewer Concerns:**

Core novelty vs. reliance on static verifiers. A central motivation is to move beyond rigid, static rule-based systems, yet the training signal in RS-DPO is still derived from existing linters (Ruff/PMD/JEP). Multiple reviewers noted that this reliance risks limiting the semantic depth of what is learned—particularly for “hard” idioms that linters cannot capture.

Easy-to-hard generalization claim not convincingly substantiated as a mechanism. While the paper shows performance gains on the hard PEP benchmark, the evidence does not clearly isolate why these gains reflect genuine easy-to-hard generalization rather than improved instruction-following or dataset-specific transfer.

Fit for venue. The contribution is primarily an applied training recipe and benchmark design for a software engineering / programming languages task. The paper does not introduce new learning methodology and the empirical contributions, while useful, are closer to a domain study than an advance in representation learning or learning theory typically emphasized at ICLR.

**Reviewer Scores:**

ZLtP: Remains at 6 (marginal accept; would not mind rejection). The rebuttal addressed most of their concerns, but the remaining limitations are broader (novelty/venue fit).
vtVc: 6 (marginal accept; would not mind rejection). Their main reservations about novelty, missing baselines, and localization weakness remain only partially resolved.
cMmU: 4 (marginal reject). They explicitly remained unconvinced that the paper demonstrates a stronger code quality analyzer, and the follow-up still does not provide decisive quantitative evidence supporting the broader claims.
TK22: 4 (marginal reject). Concerns about contradiction between motivation and linter-based feedback, and limited methodological novelty beyond known DPO effects, remain.

---

### Decision · Program_Chairs · 2026-01-26

Reject